# 10-year level, trends and socio-demographic disparities of obesity among Ghanaian adults —A systematic review and meta-analysis of observational studies

**Mustapha Titi Yussif** *, **Araba Egyirba Morrison, Reginald Adjetey Annan**

Department of Biochemistry and Biotechnology, Faculty of Science, Kwame Nkrumah University of Science and Technology, Kumasi, Ghana

* rastiti2009@hotmail.com

**Data Availability Statement:** Yes - all data is fully available within the Manuscript and Supporting Information files.

## Abstract

The double burden of malnutrition has assumed severer forms in Low and Middle Income Countries (LMICs) arising from sharper increases in prevalence rates of overweight and obesity in these countries compared to higher income countries. Considering that LMICs already have fragile health systems, the rising obesity levels may just be a ticking time bomb requiring expeditious implementation of priority actions by all global and national actors to prevent an explosion of cardiovascular disease related deaths. The aim of this systematic review and meta-analysis was to provide a current estimate of obesity and overweight prevalence among Ghanaian adults and assess socio-demographic disparities following the PRISMA guidelines. We searched Pubmed with Medline, Embase, Science direct and African Journals Online (AJOL) for studies on overweight and obesity published between 2013 and January 2023. Applying a quality effects model, pooled mean Body Mass Index (BMI) and prevalence of overweight and obesity were obtained from 42 studies conducted across all three geographical locations of Ghana with a combined sample size of 29137. From the analysis, the mean BMI of adults in Ghana was 24.7 kgm$^{-2}$ while overweight and obesity prevalence was estimated as 23.1% and 13.3% respectively. Temporal analysis showed sharper increases in overweight and obesity prevalence from 2017/2018. Mean BMI (Females: 25.3kgm$^{-2}$ vrs Males: 23.1 kgm$^{-2}$), overweight (Females: 25.9% vrs Males: 16.5%) and obesity (Females: 17.4% vrs Males: 5.5%) prevalence were higher among females than males. Gender differences in mean BMI and obesity prevalence were both significant at p<0.001. Urban dwellers had higher mean BMI than their rural counterparts (24.9kgm$^{-2}$ vrs 24.4kgm$^{-2}$). Overweight (27.6% vrs 18.2%) and obesity (17.3% vrs 11.0%) prevalence were also higher in urban areas than in rural areas. Body weight indicators for the various geographical areas of Ghana were; southern sector: 25.4kgm$^{-2}$, 28.9% and 15.4%, middle sector: 24.8kgm$^{-2}$, 26.4% and 16.2% and northern sector: 24.2kgm$^{-2}$, 15.4% and 8.5% for mean BMI, overweight and obesity prevalence respectively. The southern part of Ghana was similar to the middle part in terms of mean BMI, overweight and obesity but higher than the northern part. We conclude that overweight and obesity prevalence in Ghana has risen to high levels in recent years with women and urban dwellers

**Funding:** The authors received no specific funding for this work.

**Competing interests:** The authors have declared that no competing interests exist.

disproportionately more affected. There is a possible implication for increased cardiovascular diseases and a generally poor quality of life for the people. Evidence-based public health interventions are needed to reverse the current situation.

## Introduction

According to global estimates, 17.9 million people died from cardiovascular diseases (CVDs) in 2019, accounting for 32% of all deaths worldwide making CVDs the leading cause of death worldwide with low and middle-income countries (LMICs) accounting for more than 75 percent of these CVD mortalities [1]. Sub Saharan Africa (SSA) has seen more than 50% increases in the number of CVD deaths in the past three decades [2] and given the growing burden of CVDs in SSA, projections indicate that in a few more decades, CVDs and other Non-Communicable Diseases (NCDs) will replace communicable diseases as the leading cause of death in this region [3].

The epidemiology of CVDs has received extensive study, and there are numerous risk factors associated with it, including overweight and obesity [1, 4].

According to the International Classification of Diseases [5], Obesity can be classified separately as a disease and it has been further described by Bray et al [6] as a chronic, relapsing multifactorial disease that is a significant risk factor for not only CVD but other NCDs. It is becoming an epidemic in almost every country around the globe with global prevalence rates haven doubled between 1980 and 2014 [7].

Global efforts at curtailing the obesity menace has failed as the WHO target of halting the rise in obesity at the 2010 levels by 2025 has been missed by countries as global prevalence is likely to further double with one billion people likely to be obese worldwide by 2030 according to the World Obesity Federation [8]. Figures from the 2022 World Obesity Atlas [8] further indicates that LMICs will have the number of people with obesity more than doubling and even tripling in low income countries by 2030. This makes LMICs which are woefully unprepared and ill-equipped to deal with obesity and its effects, home to majority of obese people in the world. These staggering figures of obesity in LMICs will worsen the already precarious CVD situation and together with the concurrent undernutrition situation contribute to what is now described as severe double burden of malnutrition [9].

The drivers and determinants of this ever rising overweight and obesity epidemic in LMICs are wide-ranged, complex and cross-linked however, there is general consensus around two main contributory factors to the rising overweight and obesity rates viz the nutrition transition and sedentary lifestyles [10]. First, there has been a considerable change in dietary habits over the past three decades where there is an increased consumption of animal source foods, caloric sweeteners and ultra-processed foods that are high in fat, sugar, oils and a shift away from traditional diets rich in cereals, whole grains, and pulses in what has been described as the nutrition transition [11–13]. This nutrition transition has largely been attributed to income growth and accelerating urbanization rates which facilitates a better financial and geographical access to a growing modern food retail sector as described by Pokin and Gordon-Larsen [14]. The second contributory factor is the shift from physically demanding domestic and agricultural work to less strenuous tasks, using less energy for daily tasks, chores, and transportation [12, 15].

In Ghana, overweight and obesity have been identified to have substantial health and economic impacts in terms of life expectancy (LE), quality-adjusted life years (QALYs) and

lifetime costs in the adult population. Lartey et al [16] estimate that a total of 267 859 Years of Life Lost, 247 799 QALYs, and an additional cost of US$82 million were lost as a result of overweight and obesity during a 50-year period in the entire Ghanaian population aged 50 years, with 64% of those costs falling on the government's National Health Insurance Scheme.

Meanwhile, the prevalence of overweight and obesity continues to rise in Ghana over the years. The Ghana Demographic and Health Survey (GDHS) reported an increasing trend in overweight/obesity among women from 25% in 2003 to 40% in 2014 and prevalence among men to be 16% in 2014 [17]. From the review of the literature, the most recent estimate of the prevalence of overweight and obesity in Ghana was a meta-analysis conducted by Ofori-Asenso et al [18] which combined studies that have been published up to March 2016 in which 43% of adult Ghanaians were found to be either overweight or obese. Although this study is more recent than the GDHS data, it is almost 10 years old and may not reflect the current situation of overweight and obesity in Ghana based on which the needed public health interventions should be taken.

We set out therefore, in this review to evaluate the prevalence levels of obesity in Ghana within the past 10 years and to examine any disparities based on socio-demographic factors. In addition we sought to as well identify any existential trends in the prevalence of obesity among adult Ghanaians over the last ten years.

## Methods

This systematic review was conducted in accordance with the Preferred Reporting Items for Systematic Reviews and Meta-Analyses (PRISMA) [19] and the Conducting Systematic Reviews and Meta-Analyses of Observational Studies of Etiology (COSMOS-E) [20] guidelines.

### Search strategy

Pubmed including Medline, ScienceDirect, Embase and African Journal Online (AJOL) were systematically searched to retrieve primary literature to evaluate the current levels of overweight and obesity in Ghana. References from the articles which were obtained from the electronic database search were additionally searched as a way of capturing publications that might have been missed during the initial electronic database search. Peer reviewed publications written in English language and published over the last 10 years (2013–2023) were considered for this review.

The main search terms that were used to retrieve articles for the review were "Obesity", "Overweight", "adiposity", "and anthropometry"," Body Mass Index" and were used in combination with "Prevalence", "Ghana", and "Ghanaian" as additional terms. All database searches were conducted between October 2022 and January 2023 using the above stated search terms and the results refined using the eligibility criteria developed. All search results were saved in each of the electronic databases and exported to the Covidence software for management.

### Inclusion and exclusion criteria

**Inclusion.**  Inclusion for this review considered observational studies that had adult study populations in Ghana (aged 18 years and above). Where the study participants were women, only non-pregnant women were included irrespective of their parity (parous or nulliparous). We also included studies with the outcomes of interest irrespective of the metabolic status of the study participants.

The primary and secondary outcomes of interest for this review which constituted the cardinal basis of inclusion were publications that measured the prevalence of overweight and

obesity which were determined based on Body Mass Index (BMI) and calculated as weight in kilograms divided by height in meters squared. Studies were included if they defined a BMI of 25–29.9kgm$^{-2}$ and BMI $\geq$ 30kgm$^{-2}$ as overweight and obesity respectively in accordance with WHO standards [21]. Studies that did not report prevalence of overweight and obesity but presented enough data for prevalence to be calculated were included and studies that reported only the mean BMI were as well included.

**Exclusion.** Studies were excluded from the review if the study participants were children or adolescents (aged less than 18 years old) or pregnant women and also if the studies evaluated obesity based on waist circumference and other parameters other than BMI.

We excluded studies that measured overweight and obesity together; overweight/ obesity = BMI $\geq$ 25kgm as we could not determine separate prevalence rates for overweight and obesity.

**Data extraction and quality assessment.** Data extraction was independently done by MTY and AEM who are the primary investigators and the information from each of the included studies was documented using forms generated from the Covidence software and differences resolved by consensus. A pilot extraction was initially conducted by the authors using five papers after which the necessary amendments or updates were done based on sound clinical principles.

We extracted data on study characteristics such as authors' names and year of publication, year of data collection, study location, study population, study design and key socio-demographic characteristics. For study location, we regrouped the 10 regions of Ghana into three geographical zones namely Northern (Upper East, Upper West and Northern regions), Middle (Ashanti and Brong Ahafo regions) and Southern (Greater Accra, Central, Volta, Eastern and Western regions). Level of education was classified as 'High' if study participants attained an educational status of between high school/O'level and tertiary whiles an educational status lower than high school or no education was classified as 'Low'.

For quantitative information, the mean age of the study population, mean BMI and the proportions or prevalence rates of the outcome measurements in each study were recorded noting the standard deviations and 95% confidence intervals as appropriate.

As all the included studies were observational studies and cross sectional by design, critical appraisal and risk of bias assessment to determine the methodological quality was done using the AXIS tool [22]. The use of this tool provided the opportunity to assess each individual aspect of study design of the included studies to give an overall assessment of the quality of the study thus providing the opportunity to calculate a quality index (Qi) for each study for the application of the quality effects model as proposed by Doi and Thalib (2008) [23]. Quality assessment and data extraction was independently done by the investigators and differences resolved by consensus.

## Analysis

Meta-analysis was performed using MetaXL version 5.3 and Open meta-Analyst (OMA)

A quality effects model meta-analysis was applied to explore the effects of quality of the included studies on the prevalence of overweight and obesity and the mean BMI. This manages methodological heterogeneity within the studies by combining the heterogeneity effects in the overall analysis [23].

Data was largely treated as proportions in the Meta-analysis to obtain pooled prevalence for overweight and obesity. To determine the pooled mean body weight of the various study populations, we evaluated BMI as a continuous variable in the meta-analysis where the standard deviations (SDs) were used together with the sample sizes to compute the weight given to each

study. In studies where SDs were not provided we calculated them using the given sample size and 95% confidence intervals using the formula $SD = \frac{\sqrt{N} \times (Upper\ Limit - Lower\ Limit)}{3.92}$ [24].

Forest plots were used to display the results of individual studies and the synthesis. Tests for heterogeneity were performed using the Cochrane's Q statistic test and $I^2$ statistic to evaluate the percentage of residual variation attributed to heterogeneity. We then performed a subgroup analysis to explore the cause of the heterogeneity [25] and subsequently used meta-regression techniques to test for differences within sub-groups to reduce the possibility of false positive results [26]. Sub group analysis enabled us to determine the extent to which socio-demographic factors such as gender, age (emerging adults and aging adults), geographical location (regions or urban/rural) and educational level as well as diabetes status influenced the pooled effects on the outcome variables. The sub group analysis was also used to assess the socio-demographic disparities in the prevalence of overweight and obesity in the general population.

Robustness of the synthesised results was done through a leave-one-out sensitivity analysis and publication bias was assessed by using Doi plots to visualize asymmetry and Luis Furuya-Kanamori (LFK) index to quantify asymmetry of study effects in the Doi plots [27].

## Results

### General characteristics of included studies

Electronic search from all the four databases used for this review yielded 3,984 publications. Due to the use of multiple databases, 329 duplicate publications were identified and excluded whilst secondary search of reference lists led to the addition of 10 more articles. Upon title and abstract screening of the 3665 articles remaining, 3574 were excluded because they were irrelevant to the review. 91 articles were deemed relevant and thus taken through full-text reading to assess their eligibility for inclusion. Full-text reading resulted in a further exclusion of 49 articles for various reasons as detailed in the PRISMA flow chart (Fig 1). Effectively, 42 articles satisfied the inclusion criteria and were thus assessed for methodological quality.

The quality of all the 42 studies was assessed to be good as shown on Table 1 and thus included in the meta-analysis. All the studies had clear aims and objectives and used the appropriate study designs and methods and adopted samples that were representative of the study population.

All the 42 studies included in the meta-analysis were cross sectional studies published between 2013 and 2023 (Table 2) and conducted across only seven regions out of the ten regions of Ghana; Greater Accra (n = 9), Volta (n = 3), Ashanti (n = 10), Brong Ahafo (n = 1), Northern (n = 10), Upper East (n = 3) and Western region (n = 1) and nationally representative samples were used in 5 Studies.

The included studies had study participants ranging between 36 and 4337 and a pooled sample size of 29137 participants used for the meta-analysis with males and females contributing 39% and 61% respectively.

### Population mean BMI

Twenty nine (n = 29) studies with a total sample size of 16,846 reported on mean BMI among adults from all the three geographical locations in the country. The mean BMI ranged from 20.6Kgm$^{-2}$ [28] to 30.5kgm$^{-2}$ [29]. The pooled mean BMI among adults in Ghana was 24.7 kgm$^{-2}$ (95%CI: 24.0–25.4) with little to no statistical heterogeneity among the included studies implying the effect sizes are consistent across the studies ($I^2$ = 0%, p = 0.79) as indicated in the forest plot in Fig 2.

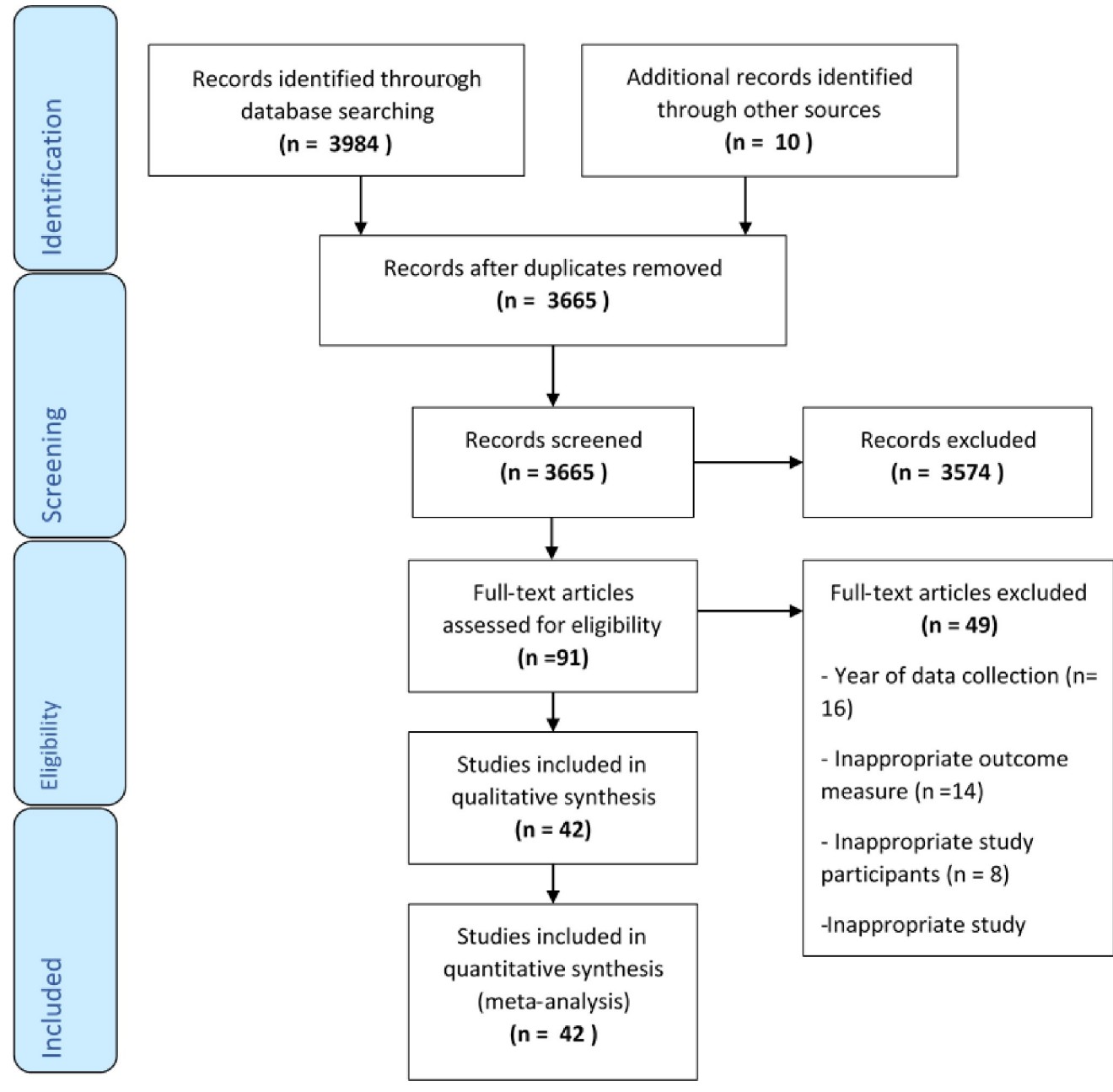

**Fig 1. PRISMA diagram of the literature search process.**

Sensitivity analysis was not conducted due to the absence of statistical heterogeneity however, publication bias was assessed. Doi plots in Fig 3 indicated no asymmetry with an LFK index of 0.9 indicating no publication bias.

In subgroup analysis, pooled data from 23 studies was used to estimate the mean BMI for males in Ghana to be 23.1 kgm$^{-2}$(95%CI: 21.7–24.4, I$^2$ = 0%, p = 0.86) whiles the mean BMI for females in Ghana was 25.3kgm$^{-2}$ (95%CI: 24.5–26.1, I$^2$ = 0%, p = 0.87) estimated from 22 studies. Females have been found to have a significantly (p = 0.001) higher mean BMI compared to males.

**Table 1. Risk of Bias Assessment of included studies.**

| | Eric Kwasi Ofori et al., 2019 | Addae et al., 2022 | York et al., 2021 | Kushitor et al., 2020 | Suara et al., 2019 | Addo et al., 2015 | Appiah et al., 2020 | Dake et al., 2016 | Mogre et al., 2015 | Lartey et al., 2019 | Obirikorang et al.,2016 | Ofori & Angmorterh, 2019 | Agongo et al. 2018 | Agyemang-Yeboah et al., 2019 |
|---|---|---|---|---|---|---|---|---|---|---|---|---|---|---|
| **Introduction** | | | | | | | | | | | | | | |
| 1. Were the aims/objectives of the study clear? | YES | YES | YES | YES | YES | YES | YES | YES | YES | YES | YES | YES | YES | YES |
| **Methods** | | | | | | | | | | | | | | |
| 2. Was the study design appropriate for the stated aim(s)? | YES | YES | YES | YES | YES | YES | YES | YES | YES | YES | YES | YES | YES | YES |
| 3. Was the sample size justified? | YES | YES | YES | NO | YES | YES | NO | NO | NO | NO | NO | YES | NO | YES |
| 4. Was the target/reference population clearly defined? (Is it clear who the research was about? | YES | YES | YES | YES | YES | YES | YES | YES | YES | YES | YES | YES | YES | YES |
| 5. Was the sample frame taken from an appropriate population base so that it closely represented the target/reference population under investigation? | YES | YES | YES | YES | YES | YES | YES | YES | YES | NS | YES | YES | NS | YES |
| 6. Was the selection process likely to select subjects/participants that were representative of the target/reference population under investigation? | YES | YES | NO | YES | ND | YES | NO | YES | YES | ND | NO | YES | ND | YES |

*(Continued)*

**Table 1.** (Continued)

| Question | Eric Kwasi Ofori et al., 2019 | Addae et al., 2022 | York et al., 2021 | Kushitor et al., 2020 | Suara et al., 2019 | Addo et al., 2015 | Appiah et al., 2020 | Dake et al., 2016 | Mogre et al., 2015 | Lartey et al., 2019 | Obirikorang et al.,2016 | Ofori & Angmorterh, 2019 | Agongo et al. 2018 | Agyemang-Yeboah et al., 2019 |
|---|---|---|---|---|---|---|---|---|---|---|---|---|---|---|
| 7. Were measures undertaken to address and categorize non-responders? | ND | ND | NO | NO | ND | ND | ND | ND | ND | ND | ND | ND | ND | ND |
| 8. Were the risk factor and outcome variables measured appropriate to the aims of the study? | YES | YES | YES | YES | YES | YES | YES | YES | YES | YES | YES | YES | YES | YES |
| 9. Were the risk factor and outcome variables measured correctly using instruments/measurements that had been trialled, piloted or published previously? | YES | YES | YES | NS | YES | YES | YES | YES | YES | YES | YES | YES | YES | YES |
| 10. Is it clear what was used to determined statistical significance and/or precision estimates? (e.g., p values, CIs) | YES | YES | YES | YES | YES | YES | YES | YES | YES | YES | YES | YES | NO | YES |
| 11. Were the methods (including statistical methods) sufficiently described to enable them to be repeated? | YES | YES | YES | YES | NO | YES | YES | YES | YES | NO | YES | YES | NO | NO |

**Results**

(*Continued*)

Table 1. (Continued)

| | Eric Kwasi Ofori et al., 2019 | Addae et al., 2022 | York et al., 2021 | Kushitor et al., 2020 | Suara et al., 2019 | Addo et al., 2015 | Appiah et al., 2020 | Dake et al., 2016 | Mogre et al., 2015 | Lartey et al., 2019 | Obirikorang et al.,2016 | Ofori & Angmorterh, 2019 | Agongo et al. 2018 | Agyemang-Yeboah et al., 2019 |
|---|---|---|---|---|---|---|---|---|---|---|---|---|---|---|
| 12. Were the basic data adequately described? | YES | YES | YES | YES | YES | YES | YES | YES | YES | YES | YES | YES | YES | YES |
| 13. Does the response rate raise concerns about non-response bias? | NO | NO | NO | NS | NS | NS | NO | NO | NO | NS | NS | NO | NS | NS |
| 14. If appropriate, was information about non-responders described? | NO | NO | YES | NO | NO | NO | NO | NO | NO | NO | NO | NO | NO | NO |
| 15. Were the results internally consistent? | YES | YES | YES | YES | YES | YES | YES | YES | YES | YES | YES | YES | YES | YES |
| 16. Were the results for the analyses described in the methods, presented? | YES | YES | YES | YES | YES | YES | YES | YES | YES | YES | YES | YES | YES | YES |
| **Discussion** | | | | | | | | | | | | | | |
| 17. Were the authors' discussions and conclusions justified by the results? | YES | YES | YES | YES | YES | YES | YES | YES | YES | YES | YES | YES | YES | YES |
| 18. Were the limitations of the study discussed? | NO | YES | YES | YES | YES | YES | NO | YES | YES | YES | YES | NO | YES | NO |
| **Others** | | | | | | | | | | | | | | |

(*Continued*)

**Table 1.** (Continued)

| | Eric Kwasi Ofori et al., 2019 | Addae et al., 2022 | York et al., 2021 | Kushitor et al., 2020 | Suara et al., 2019 | Addo et al., 2015 | Appiah et al., 2020 | Dake et al., 2016 | Mogre et al., 2015 | Lartey et al., 2019 | Obirikorang et al.,2016 | Ofori & Angmorterh, 2019 | Agongo et al. 2018 | Agyemang-Yeboah et al., 2019 |
|---|---|---|---|---|---|---|---|---|---|---|---|---|---|---|
| 19. Were there any funding sources or conflicts of interest that may affect the authors' interpretation of the results | NDis | NO | NO | NDis | NO | NDis | NO | NO | NO | NO | NDis | NDis | NO | NO |
| 20. Was ethical approval or consent of participants attained? | YES | YES | YES | YES | YES | YES | YES | YES | YES | YES | YES | YES | YES | YES |
| | Mogre et al., 2014 | Agyemang et al., 2016 | Ramsay et al., 2018 | Tuoyire et al., 2018 | Li et al., 2020 | Stringghini et al., 2016 | Obirikorang et al., 2015 | Yeboah et al., 2018 | Chauwa et al., 2020 | Abagre et al., 2022 | Osei-Yeboah et al., 2018 | Asare-Anane et al. 2015 | Mogre et al., 2015 | Mogre et al., 2014a |
| **Introduction** | | | | | | | | | | | | | | |
| 1. Were the aims/objectives of the study clear? | YES | YES | YES | YES | YES | YES | YES | YES | YES | YES | YES | YES | YES | YES |
| **Methods** | | | | | | | | | | | | | | |
| 2. Was the study design appropriate for the stated aim(s)? | YES | YES | YES | YES | YES | YES | YES | YES | YES | YES | YES | YES | YES | YES |
| 3. Was the sample size justified? | NO | NO | NO | NO | YES | YES | NO | NO | NO | YES | YES | NO | NO | NO |
| 4. Was the target/reference population clearly defined? (Is it clear who the research was about? | YES | YES | ND | YES | YES | YES | YES | YES | YES | YES | YES | YES | YES | YES |

(*Continued*)

**Table 1.** (Continued)

| | Eric Kwasi Ofori et al., 2019 | Addae et al., 2022 | York et al., 2021 | Kushitor et al., 2020 | Suara et al., 2019 | Addo et al., 2015 | Appiah et al., 2020 | Dake et al., 2016 | Mogre et al., 2015 | Lartey et al., 2019 | Obirikorang et al.,2016 | Ofori & Angmorterh, 2019 | Agongo et al. 2018 | Agyemang-Yeboah et al., 2019 |
|---|---|---|---|---|---|---|---|---|---|---|---|---|---|---|
| 5. Was the sample frame taken from an appropriate population base so that it closely represented the target/reference population under investigation? | YES | YES | ND | NO | YES | YES | YES | YES | ND | YES | YES | YES | NS | YES |
| 6. Was the selection process likely to select subjects/participants that were representative of the target/reference population under investigation? | ND | ND | ND | NO | YES | YES | YES | YES | ND | ND | ND | YES | ND | YES |
| 7. Were measures undertaken to address and categorize non-responders? | ND | NO | ND | NO | ND | ND | ND | ND | ND | ND | ND | ND | ND | ND |
| 8. Were the risk factor and outcome variables measured appropriate to the aims of the study? | YES | YES | YES | YES | YES | YES | YES | YES | YES | YES | YES | YES | YES | YES |

(Continued)

**Table 1.** (Continued)

| | Eric Kwasi Ofori et al., 2019 | Addae et al., 2022 | York et al., 2021 | Kushitor et al., 2020 | Suara et al., 2019 | Addo et al., 2015 | Appiah et al., 2020 | Dake et al., 2016 | Mogre et al., 2015 | Lartey et al., 2019 | Obirikorang et al.,2016 | Ofori & Angmorterh, 2019 | Agongo et al. 2018 | Agyemang-Yeboah et al., 2019 |
|---|---|---|---|---|---|---|---|---|---|---|---|---|---|---|
| 9. Were the risk factor and outcome variables measured correctly using instruments/measurements that had been trialled, piloted or published previously? | YES | YES | YES | YES | YES | YES | YES | YES | YES | YES | YES | YES | YES | YES |
| 10. Is it clear what was used to determined statistical significance and/or precision estimates? (e.g., p values, CIs) | NO | NO | NO | NO | YES | YES | YES | YES | NO | YES | YES | YES | YES | YES |
| 11. Were the methods (including statistical methods) sufficiently described to enable them to be repeated? | NO | NO | NO | NO | YES | YES | YES | YES | NO | NO | YES | YES | NO | NO |
| **Results** | | | | | | | | | | | | | | |
| 12. Were the basic data adequately described? | YES | YES | YES | YES | YES | YES | YES | YES | YES | YES | YES | YES | YES | YES |
| 13. Does the response rate raise concerns about non-response bias? | NO | NO | NS | NS | NS | NS | NS | NS | NS | NS | NS | NS | NS | NS |

*(Continued)*

**Table 1.** (Continued)

| | Eric Kwasi Ofori et al., 2019 | Addae et al., 2022 | York et al., 2021 | Kushitor et al., 2020 | Suara et al., 2019 | Addo et al., 2015 | Appiah et al., 2020 | Dake et al., 2016 | Mogre et al., 2015 | Lartey et al., 2019 | Obirikorang et al., 2016 | Ofori & Angmorterh, 2019 | Agongo et al. 2018 | Agyemang-Yeboah et al., 2019 |
|---|---|---|---|---|---|---|---|---|---|---|---|---|---|---|
| 14. If appropriate, was information about non-responders described? | NO | YES | NO | NO | NO | NO | NO | NO | NO | NO | NO | NO | NO | NO |
| 15. Were the results internally consistent? | YES | YES | YES | YES | YES | YES | YES | YES | YES | YES | YES | YES | YES | YES |
| 16. Were the results for the analyses described in the methods, presented? | YES | YES | YES | YES | YES | YES | YES | YES | YES | YES | YES | YES | YES | YES |
| **Discussion** | | | | | | | | | | | | | | |
| 17. Were the authors' discussions and conclusions justified by the results? | YES | YES | YES | YES | YES | YES | YES | YES | YES | YES | YES | YES | YES | YES |
| 18. Were the limitations of the study discussed? | YES | YES | YES | YES | YES | YES | YES | YES | NO | YES | NO | NO | YES | NO |
| **Others** | | | | | | | | | | | | | | |
| 19. Were there any funding sources or conflicts of interest that may affect the authors' interpretation of the results | NO | NO | NO | NO | NO | NDis | NO | NO | NO | NO | NO | NO | NO | NO |
| 20. Was ethical approval or consent of participants attained? | YES | YES | YES | YES | YES | YES | YES | YES | YES | YES | YES | YES | YES | YES |

*(Continued)*

**Table 1.** (Continued)

| | Eric Kwasi Ofori et al., 2019 | Addae et al., 2022 | York et al., 2021 | Kushitor et al., 2020 | Suara et al., 2019 | Addo et al., 2015 | Appiah et al., 2020 | Dake et al., 2016 | Mogre et al., 2015 | Lartey et al., 2019 | Obirikorang et al., 2016 | Ofori & Angmorterh, 2019 | Agongo et al. 2018 | Agyemang-Yeboah et al., 2019 |
|---|---|---|---|---|---|---|---|---|---|---|---|---|---|---|
| | Doku 2017 | Anto et al., 2020 | Obirikorang et al., 2017 | Kortei et al., 2021 | Agyei et al., 2021 | Aryee et al., 2014 | Yakong et al., 2016 | Obirikorang et al., 2016 | Agbozo et al., 2018 | Duodu 2015 | Kasu et al., 2015 | Mogre et al., 2014 | Arthur et al., 2014 | Kunutsor & Powles 2014 |
| **Introduction** | | | | | | | | | | | | | | |
| 1. Were the aims/objectives of the study clear? | YES | YES | YES | YES | YES | YES | YES | YES | YES | YES | YES | YES | YES | YES |
| **Methods** | | | | | | | | | | | | | | |
| 2. Was the study design appropriate for the stated aim(s)? | YES | YES | YES | YES | YES | YES | YES | YES | YES | YES | YES | YES | YES | YES |
| 3. Was the sample size justified? | NO | NO | NO | YES | YES | NO | NO | YES | YES | NO | NO | NO | NO | NO |
| 4. Was the target/reference population clearly defined? (Is it clear who the research was about? | YES | YES | YES | YES | YES | YES | YES | YES | YES | YES | YES | YES | YES | YES |
| 5. Was the sample frame taken from an appropriate population base so that it closely represented the target/reference population under investigation? | YES | YES | YES | YES | YES | YES | YES | YES | YES | NS | YES | YES | NO | NS |

(*Continued*)

**Table 1.** (Continued)

| | Eric Kwasi Ofori et al., 2019 | Addae et al., 2022 | York et al., 2021 | Kushitor et al., 2020 | Suara et al., 2019 | Addo et al., 2015 | Appiah et al., 2020 | Dake et al., 2016 | Mogre et al., 2015 | Lartey et al., 2019 | Obirikorang et al.,2016 | Ofori & Angmorterh, 2019 | Agongo et al. 2018 | Agyemang-Yeboah et al., 2019 |
|---|---|---|---|---|---|---|---|---|---|---|---|---|---|---|
| 6. Was the selection process likely to select subjects/participants that were representative of the target/reference population under investigation? | YES | ND | YES | ND | NO | YES | ND | ND | YES | ND | YES | YES | NO | ND |
| 7. Were measures undertaken to address and categorize non-responders? | ND | ND | NO | NO | ND | ND | ND | ND | ND | ND | ND | ND | ND | ND |
| 8. Were the risk factor and outcome variables measured appropriate to the aims of the study? | YES | YES | YES | YES | YES | YES | YES | YES | YES | YES | YES | YES | YES | YES |
| 9. Were the risk factor and outcome variables measured correctly using instruments/measurements that had been trialled, piloted or published previously? | YES | YES | YES | YES | YES | YES | YES | YES | YES | YES | YES | YES | YES | YES |
| 10. Is it clear what was used to determined statistical significance and/or precision estimates? (e.g., p values, CIs) | NO | YES | YES | YES | YES | YES | YES | NO | YES | YES | YES | NO | NO | YES |

(*Continued*)

**Table 1.** (Continued)

| | Eric Kwasi Ofori et al., 2019 | Addae et al., 2022 | York et al., 2021 | Kushitor et al., 2020 | Suara et al., 2019 | Addo et al., 2015 | Appiah et al., 2020 | Dake et al., 2016 | Mogre et al., 2015 | Lartey et al., 2019 | Obirikorang et al.,2016 | Ofori & Angmorterh, 2019 | Agongo et al. 2018 | Agyemang-Yeboah et al., 2019 |
|---|---|---|---|---|---|---|---|---|---|---|---|---|---|---|
| 11. Were the methods (including statistical methods) sufficiently described to enable them to be repeated? | YES | YES | YES | YES | NO | YES | YES | YES | YES | NO | YES | YES | NO | NO |
| *Results* | | | | | | | | | | | | | | |
| 12. Were the basic data adequately described? | YES | YES | YES | YES | YES | YES | YES | YES | YES | YES | YES | YES | YES | YES |
| 13. Does the response rate raise concerns about non-response bias? | NO | NS | NO | NS | NS | NS | NS | NO | NO | NS | NS | NO | NS | NS |
| 14. If appropriate, was information about non-responders described? | NO | NO | NO | NO | NO | NO | NO | NO | NO | NO | NO | NO | NO | NO |
| 15. Were the results internally consistent? | YES | YES | YES | YES | YES | YES | YES | YES | YES | YES | YES | YES | YES | YES |
| 16. Were the results for the analyses described in the methods, presented? | YES | YES | YES | YES | YES | YES | YES | YES | YES | YES | YES | YES | YES | YES |
| *Discussion* | | | | | | | | | | | | | | |
| 17. Were the authors' discussions and conclusions justified by the results? | YES | YES | YES | YES | YES | YES | YES | YES | YES | YES | YES | YES | YES | YES |

*(Continued)*

**Table 1.** (Continued)

| | Eric Kwasi Ofori et al., 2019 | Addae et al., 2022 | York et al., 2021 | Kushitor et al., 2020 | Suara et al., 2019 | Addo et al., 2015 | Appiah et al., 2020 | Dake et al., 2016 | Mogre et al., 2015 | Lartey et al., 2019 | Obirikorang et al.,2016 | Ofori & Angmorterh, 2019 | Agongo et al. 2018 | Agyemang-Yeboah et al., 2019 |
|---|---|---|---|---|---|---|---|---|---|---|---|---|---|---|
| 18. Were the limitations of the study discussed? | YES | YES | NO | NO | YES | YES | NO | NO | YES | NO | NO | YES | NO | YES |
| 19. Were there any funding sources or conflicts of interest that may affect the authors' interpretation of the results | NO | NO | NO | NO | NO | NO | NO | NO | NO | NO | NO | NO | NO | NO |
| 20. Was ethical approval or consent of participants attained? | YES | YES | YES | YES | YES | YES | YES | YES | YES | YES | YES | YES | YES | YES |

Others

Abbreviations: ND–not described; NDis–not disclosed; NS–not stated

**Table 2. Characteristics of included studies.**

| S/N | Author & year of publication | Year of data collection | Study Location | Study design | Study Population | Mean age of study population | Setting | Age group | Religion | Poverty level |
|---|---|---|---|---|---|---|---|---|---|---|
| 1. | Eric Kwasi Ofori et al., 2019 [54] | N/A | University of Ghana/Greater Accra | institution based cross sectional study | University students | 30 ± 7.9 | urban | emerging adults | N/A | N/A |
| 2 | Addae et al., 2022 [55] | 2018 | Tamale/ Northern Region | facility based cross sectional study | post-partum mothers with children aged between 6 and 24 months | 28.0 (±5.8) | urban | emerging adults | 94.1% muslims and 5.9% christians | 47.7% households with high wealth index |
| 3 | York et al., 2021 [56] | 2015 | All regions | longitudinal cohort study | older adults aged 50 years and above | 62.2 (±9.9) | Both urban and rural (61.5% rural) | aging adults | 71.6% christians and 18.7% muslim | N/A |
| 4 | Kushitor et al., 2020 [57] | 2014 | All regions | Population based cross sectional study | women between the ages of 15–49 years | 30 (±9.9) | Both urban and rural (50.7% rural) | emerging adults | N/A | 35.6% within the rich wealth index with 20.6% middle class |
| 5 | Suara et al., 2019 [58] | 2018 | Tamale/ Northern Region | Population based cross sectional study | women between the ages of 18–59 year | N/A | urban | emerging adults | N/A | N/A |
| 6 | Addo et al., 2015 [59] | | Greater Accra | institution based cross sectional study | bank workers between the ages of 19–54 years | 3.2 (±6.9) | urban | emerging adults | N/A | N/A |
| 7 | Appiah et al., 2020 [60] | | Kumasi/ Ashanti Region | institution based cross sectional study | commercial taxi drivers 20 years and above | 41 (±8.9) | urban | emerging adults | N/A | N/A |
| 8 | Dake et al., 2016 [61] | 2013 | Accra/ Greater Accra | Population based cross sectional study | adults aged 15–49 years | 31.5 (±10.46) | urban | emerging adults | N/A | N/A |
| 9 | Mogre et al., 2015 [62] | 2014 | Tamale/ Northern Region | facility based cross sectional study | type 2 diabetes melitus patients | 48.48 ± 11.72 | urban | both emerging and aging adults | N/A | N/A |
| 10 | Lartey et al., 2019 [63] | 2015 | All regions | longitudinal cohort study | WHO-SAGE Wave 2 participants. Older adults aged 50 years and above | N/A | Urban and rural | | N/A | N/A |
| 11 | Obirikorang et al.,2016 [31] | 2015 | Kumasi/ Ashanti Region | facility based cross sectional study | Newly diagnosed type 2 diabetes mellitus patients | 51.14 (± 14.45) | urban | both emerging and aging adults | N/A | 53.8% with low socio-economic income |
| 12 | Ofori & Angmorterh, 2019 [54] | N/A | Accra/Greater Accra | Population based cross sectional study | University students | 30 ± 7.9 | urban | emerging adults | N/A | N/A |

*(Continued)*

**Table 2.** (Continued)

| S/N | Author & year of publication | Year of data collection | Study Location | Study design | Study Population | Mean age of study population | Setting | Age group | Religion | Poverty level |
|---|---|---|---|---|---|---|---|---|---|---|
| 13 | Agongo et al. 2018 [32] | 2015 | Navrongo/ Upper East Region | Population based cross sectional study | adults aged 40–60 years | 51 ± 6 | rural | emerging adults | N/A | 54.03% have poor socio-economic 14status |
| 14 | Agyemang-Yeboah et al., 2019 [64] | N/A | Kumasi/ Ashanti Region | facility based cross sectional study | adult diabetes patients | 58.5 ± 9.9 | urban | both emerging and aging adults | N/A | N/A |
| 15 | Mogre et al., 2014 [36] | 2013 | Tamale/ Northern Region | facility based cross sectional study | adult diabetes patients | 67.53 ± 13.32 | urban | both emerging and aging adults | N/A | N/A |
| 16 | Agyemang et al., 2016 [65] | 2015 | Kumasi/ Ashanti Region | Population based cross sectional study | Adult Ghanaians between the ages of 25–70 years | 46.2 and 46.5 for male rural and urban dwellers respectively 46.7 and 44.7 for female rural and urban dwellers respectively | Both urban and rural | both emerging and aging adults | N/A | N/A |
| 17 | Ramsay et al., 2018 [28] | N/A | Navrongo/ Upper East Region | Population based cross sectional study | Adults between 40 and 60 years | N/A | rural | emerging adults | N/A | N/A |
| 18 | Tuoyire et al., 2018 [66] | 2014 | Tamale and Accra | Population based cross sectional study | adult women | 33 ± 9.2 | urban | | N/A | N/A |
| 19 | Li et al., 2020 [67] | 2016 | Nationally representative | Population based cross sectional study | adults aged 18 years and above | 46.9±17.2 | Both urban and rural | both emerging and aging adults | N/A | N/A |
| 20 | Stringhini et al., [68] | 2016 | Nkwantakese/ Ashanti Region | Population based cross sectional study | adults aged 25 to 45 years | 34.5 (6.6) | rural | emerging adults | N/A | 35.2% are at the low wealth tertile |
| 21 | Obirikorang et al.,2015 [69] | 2013 | Kumasi and Pramso/ Ashanti Region | Population based cross sectional study | adults aged 20 years and above | 50.0 (39.0–58.0) | both rural and urban | both emerging and aging adults | N/A | 60.7% in low income class |
| 22 | Yeboah et al., 2018 [70] | 2016 | Accra/ Greater Accra | institution based cross sectional study | students aged between 20 and 30 years | 24.9 ± 2.9 | urban | emerging adults | N/A | NA |
| 23 | Chauwa et al., 2020 [71] | N/A | Kumasi/ Ashanti Region | facility based cross sectional study | out patient stroke survivors aged 18 years and above | 58.5±14.2 | urban | both emerging and aging adults | 92% Christians | N/A |
| 24 | Abagre et al., 2022 [72] | 2021 | Berekum and Dormaa/Bono Region | facility based cross sectional study | diabetes patients between the ages of 30–79 years old | 58.84 ± 11.49 | rural | both emerging and aging adults | N/A | N/A |

*(Continued)*

**Table 2.** (Continued)

| S/N | Author & year of publication | Year of data collection | Study Location | Study design | Study Population | Mean age of study population | Setting | Age group | Religion | Poverty level |
|-----|------|------|------|------|------|------|------|------|------|------|
| 25 | Osei-Yeboah et al., 2018 [73] | 2016 | Sefwi-Wiaso/ Western Region | institution based cross sectional study | Health care workers in the Sefwi-Wiawso municipal hospital aged between 22 and 59 | 32.1 ± 8.9 | urban | emerging adults | N/A | N/A |
| 26 | Asare-Anane et al., 2015 [74] | N/A | Tema/Greater Accra | institution based cross sectional study | Cocoa processing factory workers | 42.0 ± 8.2 and 40.3 ± 11.5 years respectively for shift and non-shift workers | urban | emerging adults | N/A | N/A |
| 27 | Mogre et al., 2015 [62] | 2014 | Tamale/ Northern Region | facility based cross sectional study | diabetes patients 20–70 years seeking care | 47.3 ± 12.73 | urban | both emerging and aging adults | N/A | N/A |
| 28 | Mogre et al., 2014 [75] | 2013 | Tamale/ Northern Region | facility based cross sectional study | diabetes patients | 56.2±12.13 | urban | both emerging and aging adults | N/A | N/A |
| 29 | Doku, 2017 [76] | 2013 | Accra/Greater Accra | institution based cross sectional | university staff | | urban | both emerging and aging adults | N/A | N/A |
| 30 | Anto et al., 2020 [77] | 2015 | Accra and Kumasi | | licensed drivers of Metromass transit buses | 44.07 ± 9.29 | urban | emerging adults | N/A | 14.3% are high income earners |
| 31 | Obirikorang et al., 2017 [78] | 2013 | Kumasi/ Ashanti Region | institution based cross sectional study | undergraduate students in a public university | 20.88±0.80 | urban | emerging adults | N/A | N/A |
| 32 | Kortei et al., 2021 [79] | 2018 | Ho/Volta region | population based cross sectional study | adults between 18 and 60 years | | urban | emerging adults | N/A | N/A |
| 33 | Agyei et al., 2022 [80] | 2020 | Accra/Greater Accra | institution based cross sectional study | undergraduate medical students between 20 and 35 years | | urban | emerging adults | 96.9% muslim | N/A |
| 34 | Aryee et al., 2014 [81] | N/A | Tamale/ Northern Region | facility based cross sectional study | nurses working in four hopsitals in Tamale | | urban | emerging adults | 55% christianity | N/A |
| 35 | Yakong et al., 2016 [82] | 2013 | Tamale/ Northern Region | population based cross sectional study | market men and women in Tamale | 35.0 ± 9.7 | urban | both emerging and aging adults | N/A | N/A |
| 36 | Obirikorang et al., 2016 [31] | N/A | Kumasi/ Ashanti Region | facility based cross sectional study | practicing nurses at 4 hospitals in kumasi | 31.55 ± 9.67 | urban | emerging adults | N/A | N/A |
| 37 | Agbozo et al., 2018 [83] | N/A | Accra/ Greater Accra | population based cross sectional study | elderly men and women between the ages of 60–70 | | urban | aging adults | 90% christian | N/A |

(*Continued*)

**Table 2.** (Continued)

| S/N | Author & year of publication | Year of data collection | Study Location | Study design | Study Population | Mean age of study population | Setting | Age group | Religion | Poverty level |
|---|---|---|---|---|---|---|---|---|---|---|
| 38 | Duodu 2015, [84] | 2015 | Hohoe/Volta Region | Institution based cross sectional study | nurses and midwives 18 years and above | 36±12.6 | both urban and rural | emerging adults | 93.6% christianity | N/A |
| 39 | Kasu et al., 2015 [85] | 2013 | Kajebi/Volta Region | Institution based cross sectional study | All health workers in health facilities in the Kadjebi District | 34.4 | rural | emerging adults | N/A | N/A |
| 40 | Mogre et al., 2014 [30] | 2013 | Tamale/Northen Region | Institution based cross sectional study | students of school of medicine and health sciences, UDS | 23.06 ± 2.77 | Urban | emerging adults | N/A | N/A |
| 41 | Arthur et al., 2014 [86] | 2013 | Kumasi/Ashanti region | Institution based cross sectional study | academic and administrative staff of KNUST, Kumasi | N/A | Urban | emerging adults | N/A | N/A |
| 42 | Kunutsor and Powles 2014 [87] | N/A | Kasena Nankana District/Upper East Region | Population based cross sectional study | adult men and women | 37.8 ±14.1 | rural | emerging adults | N/A | N/A |

Ghanaians living in urban areas have also been estimated to have a mean BMI of 24.9kgm$^{-2}$ (95%CI: 24.0–25.8, $I^2$ = 0%, p = 0.87) compared to rural dwellers with a mean BMI of 24.4kgm$^{-2}$ (95%CI: 23.3–25.5, $I^2$ = 0%, p = 0.46) following a sub group analysis that combined 20 studies and 3 studies respectively

Eight articles that reported on mean BMI were conducted in the geographical southern part of Ghana. Pooled estimates of mean BMI for this part of the country was determined to be 25.4kgm$^{-2}$ (95%CI: 23.9–26.9, $I^2$ = 0%, p = 0.97). The geographical middle part of Ghana has a mean BMI of 24.8kgm$^{-2}$ (95%CI: 23.7–25.9, $I^2$ = 0%, p = 0.75) when seven studies (with n =) were combined in the meta analysis whiles thenorthern part of the country has a mean BMI of 24.2kgm$^{-2}$ (95%CI: 23.0–25.4, $I^2$ = 0%, p = 0.30) after combining nine studies. Further sub group analysis based on age category (18-59/60+), educational level (low or high), year of data collection, and marital status showed similarities in mean BMI between subgroups as shown in Table 3.

## Current level of Overweight and socio-demographic disparities among adults in Ghana

The prevalence of overweight in Ghana is 23.1% (95%CI: 17.1–29.7%; $I^2$ = 97% p< = 0.00) as depicted in the forest plot in Fig 4. This was estimated from a meta-analysis that pooled prevalence rates from twenty nine studies with prevalence rates ranging from 4.0% (95%CI:2.4–6.1%) [30] to 69.8% (95%CI:64.5–74.8%) [31]among individual studies and a combined sample size (n) of 18,388 study participants. Sensitivity analysis was conducted using leave-one out analysis which indicated that the estimated pooled prevalence was largely affected by overweight prevalence rates among females as reported by Obirikorang et al. [31]. An assessment of publication bias using Doi plots (Fig 5) indicated no bias based on the absence of asymmetry and an LFK index of 0.81

Seventeen studies reported prevalence rates of overweight among males with a combined sample size of 5037 whereas nineteen studies with a combined sample size of 10488

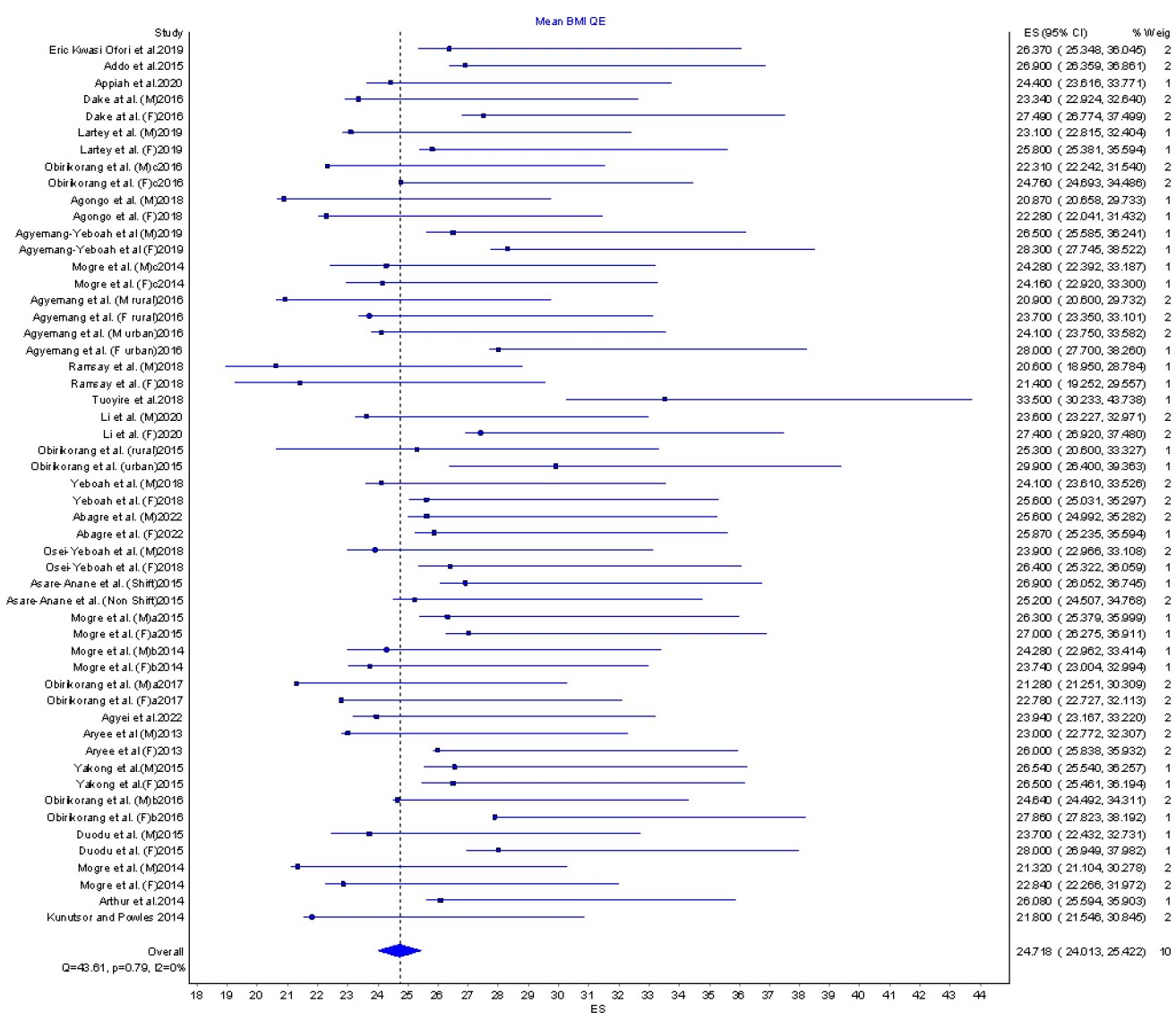

**Fig 2. Forest plot of mean BMI.**

participants reported overweight prevalence for females. The pooled overweight prevalence for males and females in Ghana was estimated as 16.5% (9.1–24.5% $I^2$ = 97%, p = 0.00) and 25.9% 19.2–32.9%; $I^2$ = 96%, p = 0.00) respectively. Prevalence of overweight for the aged (>60 years) and adults (18 – 59years) were estimated as 24.1% (22.0–26.2%, $I^2$ = 0%, p = 0.59) and 23.1% (16.1–30.5% $I^2$ = 97%, p = 0.00) after pooling prevalence rates from two studies and twenty studies with 1783 and 13373 total sample sizes respectively. A sub group analysis of overweight prevalence according to geographical location showed an overweight prevalence of 26.4% (95%CI; 12.7–41.3%, $I^2$ = 97%, p = 0.00, 6 studies, n = 2092) in the middle part of Ghana, 28.9% (95%CI; 24.3–33.7%, $I^2$ = 76%, p = 0.00, 11 studies, n = 3707) in Southern Ghana and 15.4% (95%CI; 9.7–21.5%, $I^2$ = 97%, p = 0.00, 8 studies, n = 6476) in Northern Ghana. Urban areas in Ghana have an overweight prevalence of 27.6% (95%CI; 21.1–34.4%, $I^2$ = 96%, p = 0.00) whereas an overweight prevalence of 18.2% (95%CI; 10.0–27.3%, $I^2$ = 97%, p = 0.00) was estimated for rural areas. These pooled estimates were obtained after combining

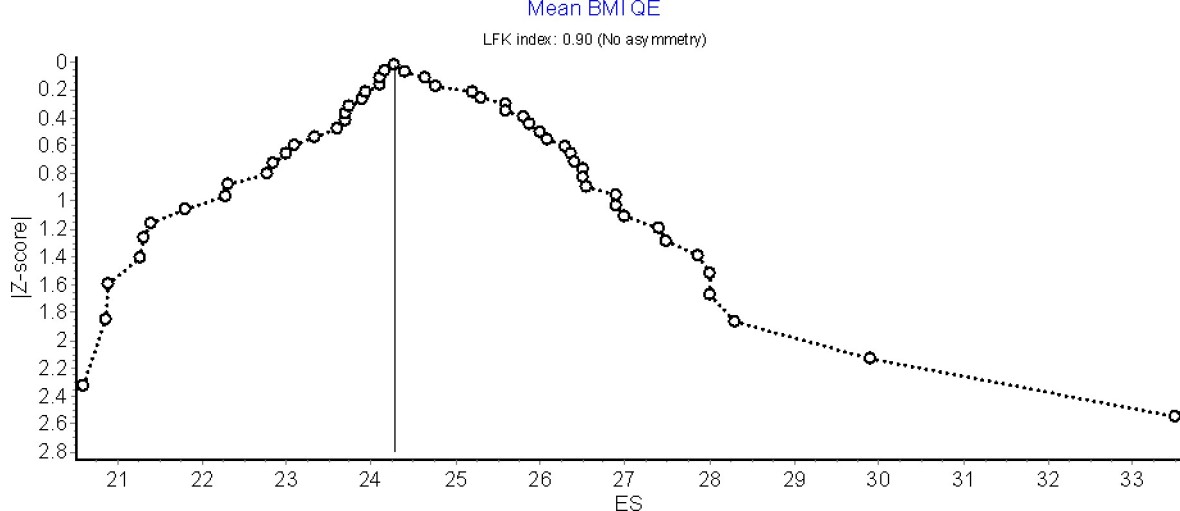

**Fig 3. Doi plot of studies reporting mean BMI in Ghana.**

**Table 3. Variation in mean BMI according to sub groups.**

| Sub-group | No. of Studies | No. of Participants | Mean BMI (Kgm$^{-2}$) | 95% Confidence Interval |
|---|---|---|---|---|
| **Gender** | | | | |
| **Male** | 23 | 5954 | 23.1 | 21.7–24.4 |
| **Female** | 22 | 8583 | 25.3 | 24.5–26.1 |
| **Age Category** | | | | |
| **Adult (18–59)** | 16 | 8789 | 24.7 | 24.0–25.4 |
| **Aged (60 +)** | 1 | 1663 | 24.4 | 20.9–27.9 |
| **Setting** | | | | |
| **Urban** | 20 | 8363 | 24.9 | 24.0–25.8 |
| **Rural** | 3 | 4283 | 24.4 | 23.3–25.5 |
| **Educational Level** | | | | |
| **High** | 9 | 2677 | 24.8 | 23.9–25.6 |
| **Low** | 8 | 6975 | 24.6 | 23.4–25.8 |
| **Year** | | | | |
| **2013/2014** | 14 | 6395 | 24.5 | 23.5–25.5 |
| **2015/2016** | 9 | 7253 | 24.9 | 23.7–26.0 |
| **2017/2018** | 1 | 2014 | 21.00 | 17.4–24.5 |
| **2019/2020** | 4 | 754 | 25.8 | 23.5–28.2 |
| **2021/2022** | 1 | 430 | 25.7 | 22.1–29.4 |
| **Location** | | | | |
| **Southern** | 8 | 1889 | 25.4 | 23.9–26.9 |
| **Middle** | 8 | 5160 | 24.8 | 23.7–25.9 |
| **Northern** | 10 | 6371 | 24.2 | 23.0–25.4 |
| **Marital status** | | | | |
| **Married** | 7 | 4078 | 24.9 | 24.1–25.8 |
| **Single** | 7 | 4460 | 24.1 | 22.5–25.7 |

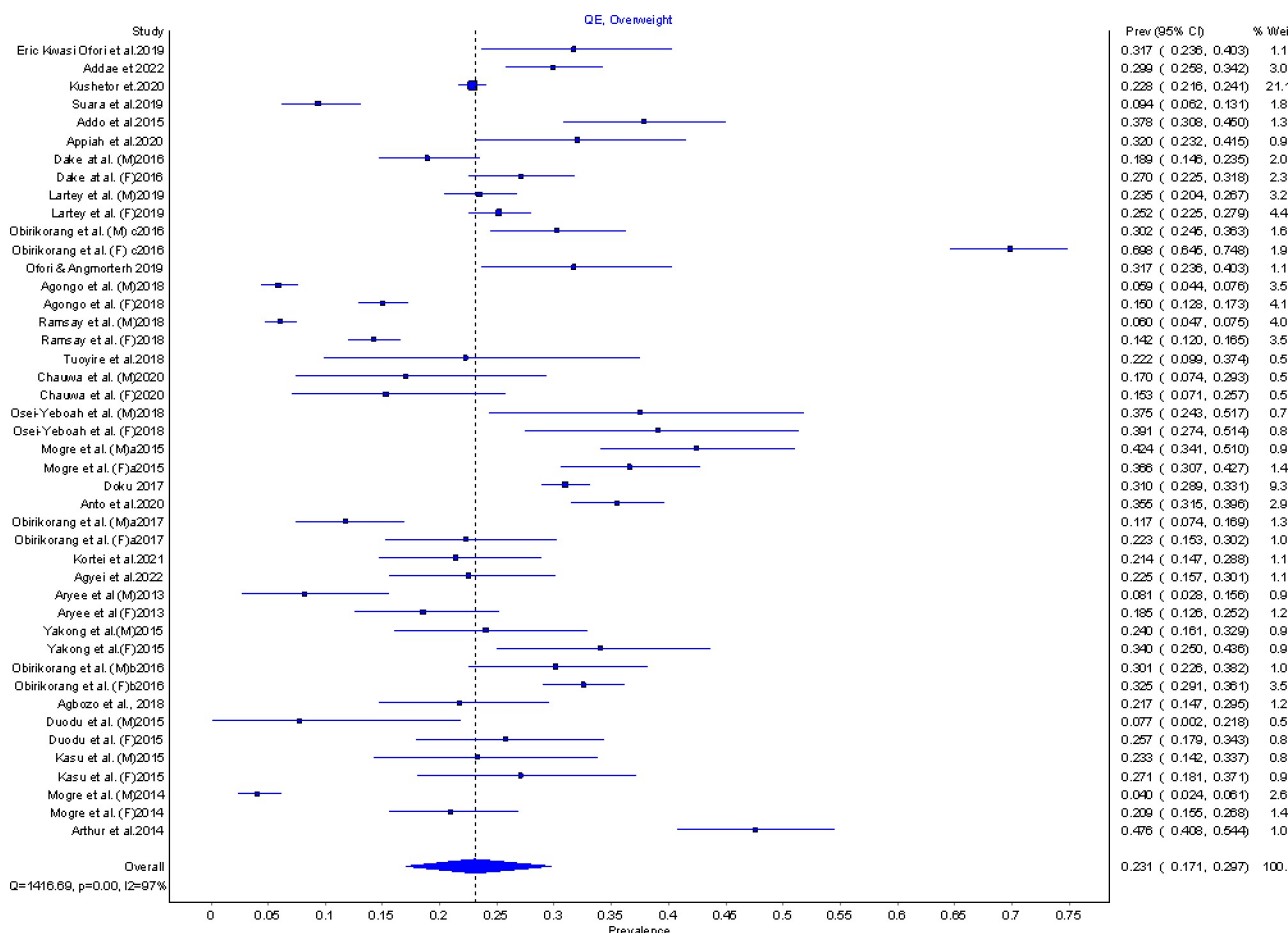

**Fig 4. Forest plot of overweight prevalence in Ghana.**

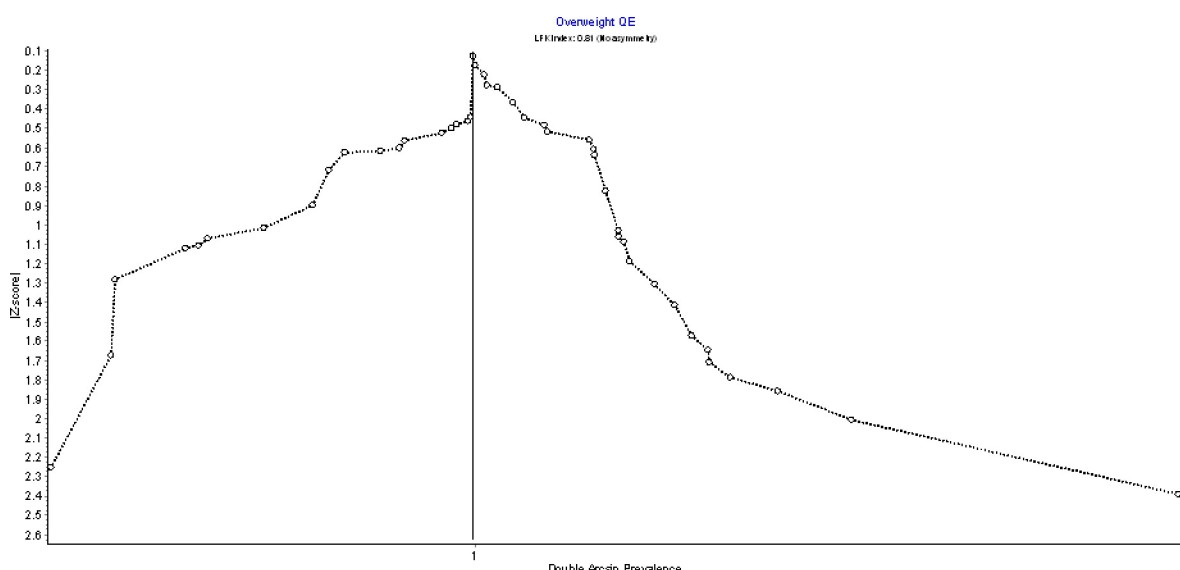

**Fig 5. Doi plot of studies reporting overweight prevalence in Ghana.**

24 studies (n = 8868) and 3 studies (n = 4011) conducted in urban and rural areas of Ghana respectively.

Overweight prevalence among Ghanaians who are married is 23.8% (95%CI; 16.5–31.4%, $I^2$ = 97%, p = 0.00, 9 studies, n = 4414) and those who are single is 20.9% (95%CI; 12.6–29.8%, $I^2$ = 95%, p = 0.00, 7 studies, n = 2713) whiles Ghanaians with 'high' educational level (senior high school to tertiary) have an overweight prevalence of 24.5% (95%CI; 17.6–31.7%, $I^2$ = 97%, p = 0.00, 14 studies, n = 5791) vis a vis 14.7% (95%CI; 6.3–24.2%, $I^2$ = 97%, p = 0.00, 2 studies, n = 2496) prevalence for Ghanaians with 'low' level of education (below senior high school/no formal education).

## Current level of obesity and socio-demographic disparities among adults in Ghana

The current level of obesity among adults in Ghana has been determined in this study to be 13.3% (95%CI: 8.3–19.3%). This was obtained through a meta-analysis that pooled prevalence data from forty two studies (with 23,931combined study participants) conducted across the country within the last 10 years. The included studies were found to vary significantly as prevalence of obesity ranged from as low as 0.9% among males in one study [32] to as high as 85.7% among females in another study [31] thus underlying the observed considerable heterogeneity ($I^2$ = 98.3, p<0.001) shown in the forest plots in Fig 6.

A sensitivity analysis using a leave-one-out analysis showed [31] to have the most impact on the pooled prevalence estimate of obesity. Doi plots (Fig 7) showed no asymmetry of the plots with an LFK index of 0.63 indicating that there was no publication bias.

Fig 8 shows the socio-demographic disparities in obesity prevalence among adults in Ghana. There was significant variability (p<0.001) in obesity prevalence based on gender as females had a significantly higher obesity prevalence of 17.4% (95% CI: 10.2–25.3, $I^2$ = 98%, p = 0.00) compared with 5.5% (95% CI: 2.6–8.8, $I^2$ = 95%, p = 0.00) among males. 23 studies (n = 12950) and 21 studies (n = 6395) respectively were used to estimate obesity prevalence for females and males.

Study setting was also identified as one of the factors that explained the disparity in obesity prevalence in the Ghanaian population. After combining 24 studies (n = 8464) that reported on obesity in urban areas in Ghana, a pooled prevalence of 17.3% (95% CI: 10.3–24.9, $I^2$ = 98%, p = 0.00) was realized whereas studies conducted in rural areas (5 studies, n = 5023) generated a pooled prevalence of 11.0% (95% CI: 4.4–18.5, $I^2$ = 99%, p = 0.00). The differences in obesity prevalence between the urban and rural areas of Ghana was found to be significant (p = 0.026).

Ghanaians with high level of education have a relatively higher prevalence of obesity of 14.9% (95%CI: 8.4–21.9, $I^2$ = 98%, p = 0.00, 15 studies, n = 6232) than those with low level of education with an obesity prevalence of 9.2% (95%CI: 1.8–18.2, $I^2$ = 99%, p = 0.00, 4 studies, n = 5562). Obesity prevalence for Adults aged 18–59 years was estimated as 13.5% (95%CI: 7.8–19.8, $I^2$ = 98%, p = 0.00, 10 studies, n = 6582) whiles adults aged 60+ years have prevalence of 14.2% (95%CI: 2.3–29.0, $I^2$ = 98%, p = 0.00, 2 studies, n = 1783). Ghanaians who are married and those who are single have 13.8% (95%CI: 7.5–20.7, $I^2$ = 98%, p = 0.00, 11 studies, n = 6210) and 13.5% (95%CI: 4.9–23.4, $I^2$ = 98%, p = 0.00, 8 studies, n = 5205) obesity prevalence respectively. There were also observed differences in obesity prevalence based on geographical location. The southern belt, middle belt and northern belts of the country have the following obesity prevalence rates respectively; 15.4% (95%CI: 9.3–22.1, $I^2$ = 91%, p = 0.00, 11 studies, n = 3707), 16.2% (95%CI: 6.5–27.3, $I^2$ = 99%, p = 0.00, 8 studies, n = 5250) and 8.5% (95%CI: 3.4–14.5, $I^2$ = 98%, p = 0.00, 10 studies, n = 7272). Meta-regression however shows

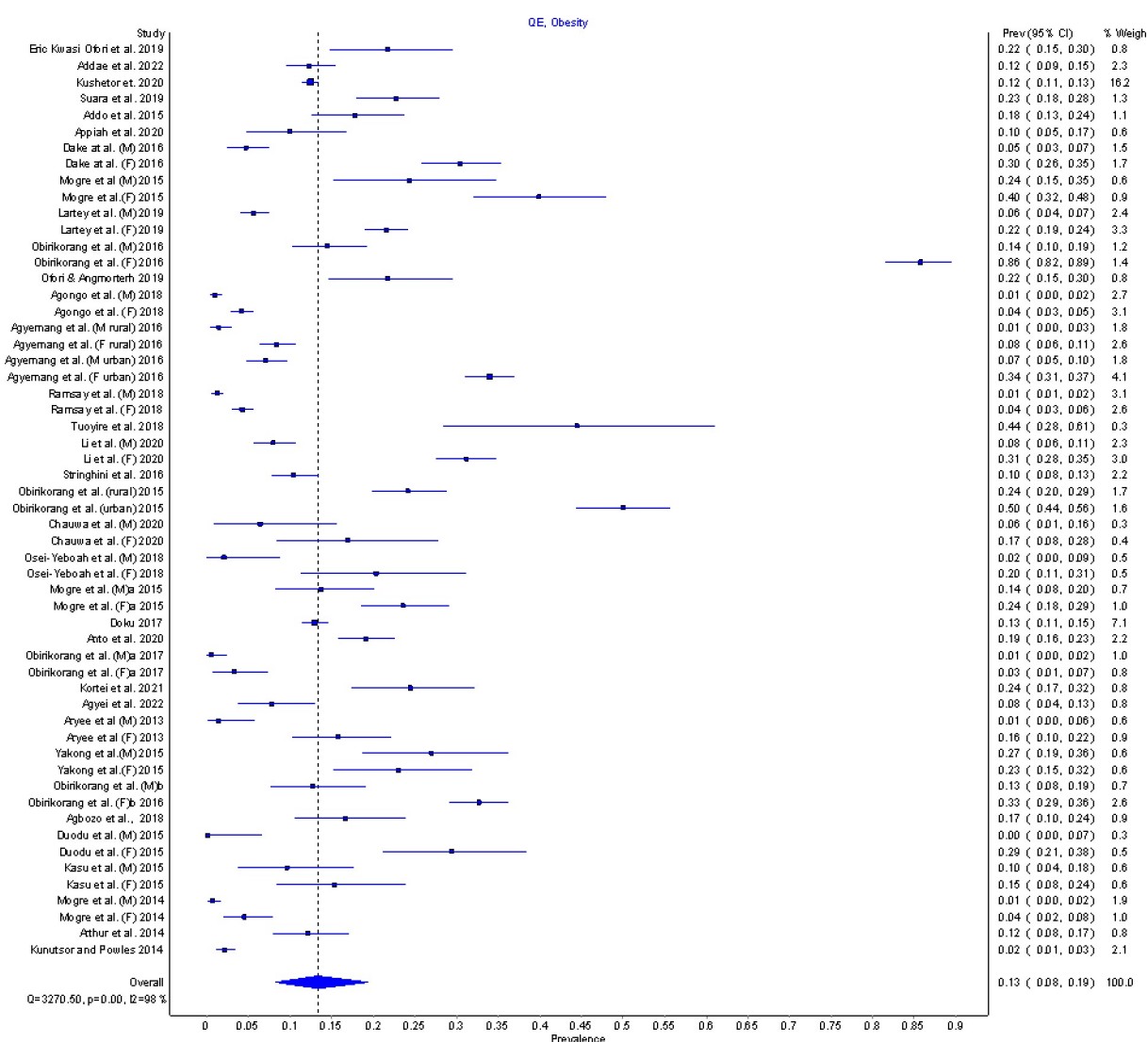

**Fig 6. Forest plot of prevalence of obesity in Ghana.**

that variability in educational level, age category and marital status did not significantly account for pooled prevalence.

## Trends in overweight and obesity prevalence in Ghana

Analysis of the trends in overweight and obesity prevalence in Ghana was done on the basis of the year of data collection in accordance with recommended practice [33] however in about 42% of the included studies, the year of publication was rather relied upon because the exact dates of data collection had not been published. The analysis was then done in two year intervals to reduce the tendency of insufficient number of studies that might be associated with using a single year window which will reduce the robustness of the meta-analysis.

For analysis of the trend in overweight prevalence, data was pooled from 12 studies (n = 10,823) in 2013/2014, 4 studies (n = 2979) in 2015/2016, 5 studies (n = 2997) in 2017/2018 and 8 studies (n = 1589) in 2019/2020. All studies published in 2021/2022 had their data

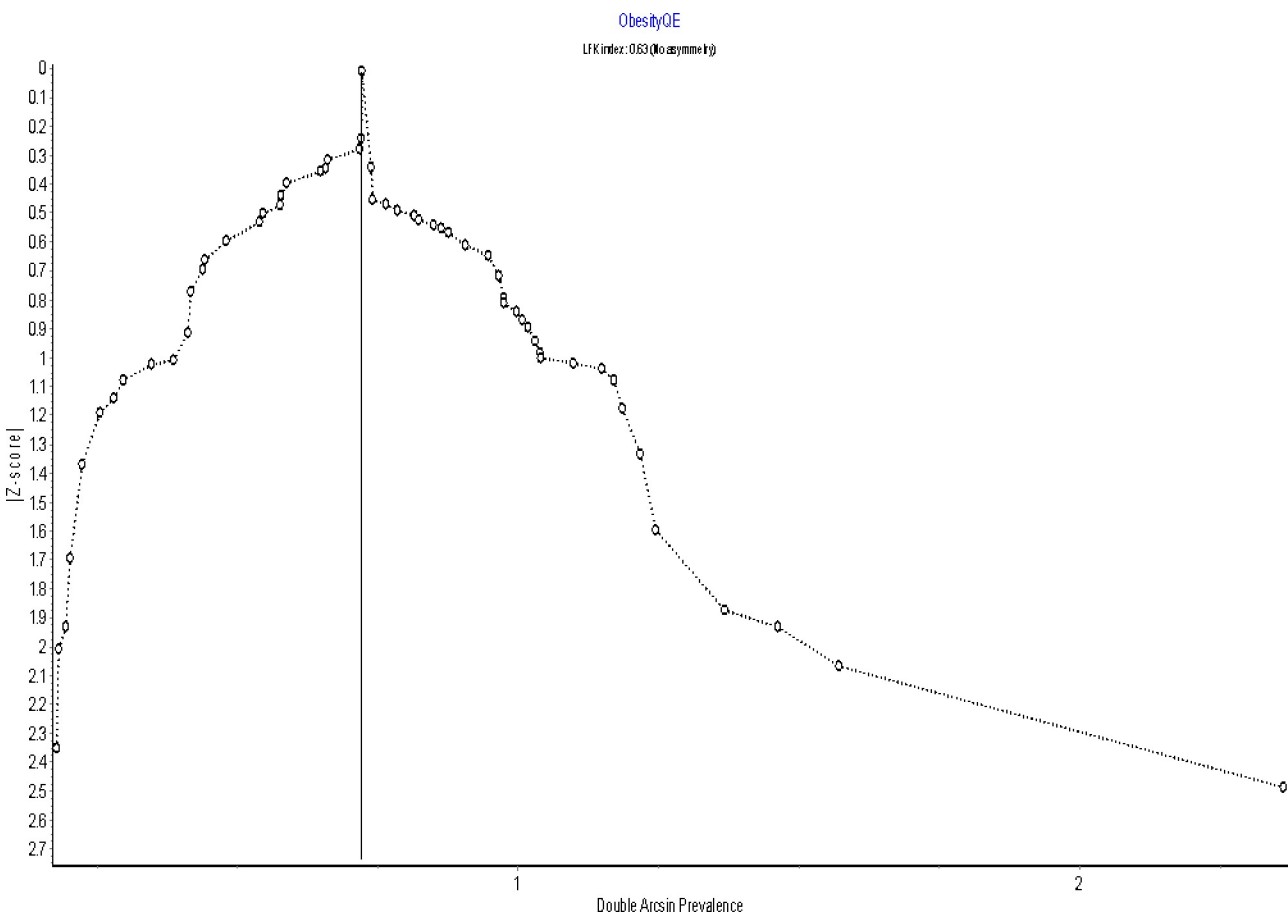

**Fig 7. Doi plot of studies reporting prevalence of obesity in Ghana.**

collected in earlier years. As shown in Fig 9, the changes in overweight prevalence as the years go by are as follows; 2013/2014–24.8% (15.5–34.6, $I^2$ = 97%.p = 0.00), 2015/2016–19.0%(7.6–32.0, $I^2$ = 98%, p = 0.00), 2017/2018–15.1%(7.2–23.9, $I^2$ = 97% p = 0.00), 2019/2020–33.2% (27.9–38.5, $I^2$ = 69%, p = 0.00). Obesity prevalence was estimated as 14.5%(4.5–26.2, $I^2$ = 95%. p = 0.00, 15 studies, n = 12291) for 2013/2014, 13.3%(5.6–22.1, $I^2$ = 99% p = 0.00, 7 studies, n = 7018) for 2015/2016, 8.6%(1.6–17.3, $I^2$ = 98% p = 0.00, 5 studies, n = 2997) for 2017/2018 and 16.8%(12.0–22.0, $I^2$ = 77% p = 0.00, 8 studies, n = 1589) for 2019/2020

## Overweight and obesity among diabetes patients in Ghana

The relationship between obesity and type 2 diabetes mellitus (T2DM) has been described to be bi-directional [34, 35] with very high probability of co-occurrence which results mostly in an increased risk of CVDs [3] and poor health related quality of life [35]. On this basis, we investigated the prevalence of obesity and overweight among people with type 2 diabetes mellitus in Ghana.

Of the included studies in this review, 1 study [31] reported overweight prevalence among diabetes patients whiles 2 studies [31, 36] reported obesity. In the study conducted by Y. Obir-korang et al. (2016) involving 543 diabetes patients, there was an overweight prevalence of 48.6% (4.6–93.6, $I^2$ = 99%, p = 0.00).

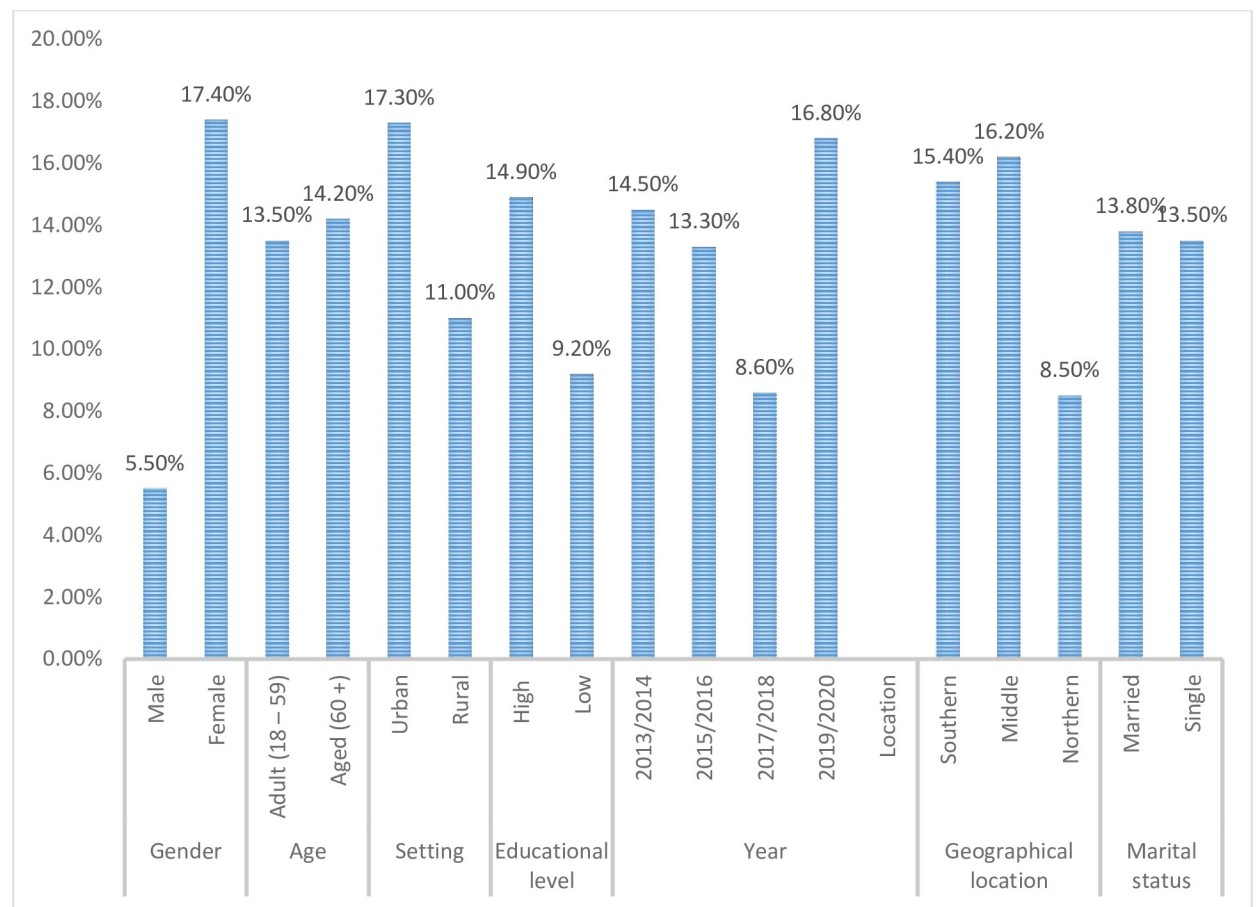

**Fig 8. Socio-demographic disparities in obesity prevalence in Ghana.**

The two studies that reported on obesity were combined in a meta-analysis to determine the pooled prevalence of obesity among T2DM patients in Ghana. Based on a combined sample size of 765, obesity prevalence was estimated as 41.7% (8.0–81.5, $I^2$ = 98, p = 0.00)

## Discussion

After systematically searching four electronic data bases and conducting a meta-analysis using 42 studies that have a combined sample size of 29137, we estimate that adults in Ghana have a mean BMI of 24.7 $kgm^{-2}$ signifying that the average adult in Ghana is of normal body weight according to the WHO classification [21] albeit at the borderline of overweight. Females, urban dwellers and people living in the geographic southern part of Ghana have exceeded the borderline and can be said to be generally overweight because their mean BMIs have exceeded 25.0 $kgm^{-2}$.

On the other hand, this study puts overweight and obesity prevalence rates among adults in Ghana at 23.1% and 13.3% respectively. These prevalence rates of overweight and obesity are similar to the findings of Ofori-Asenso et al.,[18] who conducted a meta-analysis in the recent past (2016) and estimated the prevalence of overweight and obesity among Ghanaian adults to be 25.4% and 17.1% respectively. This similarity is contrary to the expectations that overweight and obesity prevalence will continue to keep a rising trajectory in SSA countries considering that the previous study and this current one are six years apart. A plausible explanation to this

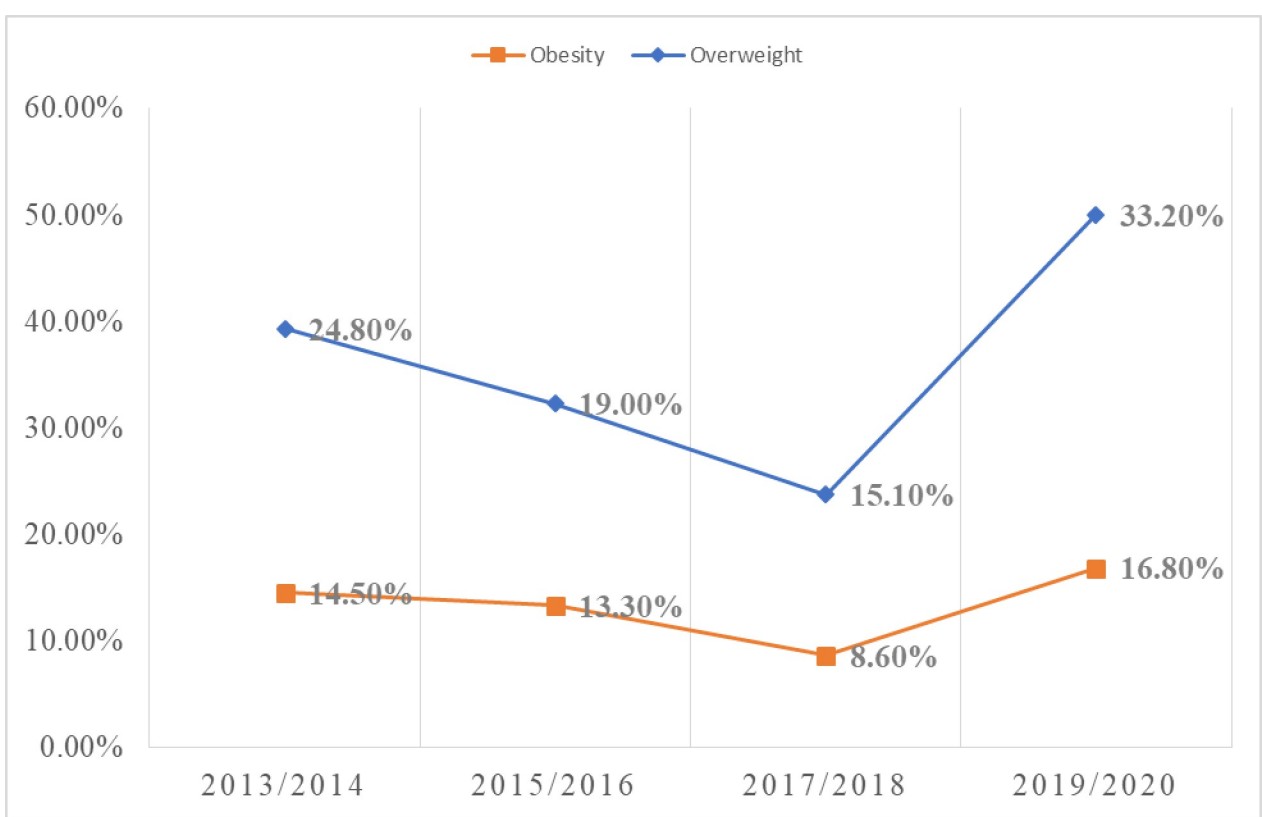

**Fig 9. Trend in overweight and obesity prevalence in Ghana.**

similarity in prevalence rates is the fact that majority of the studies that were included in the pooled estimates were conducted between 2013 and 2016. Furthermore, analysis of temporal changes in overweight and obesity in this review indicates that the earlier years, 2013/2014 and 2015/2016 have prevalence rates similar to the estimates of Ofori-Asenso et al. (2016) whiles sharper rise in prevalence rates were observed in the later years (2019/2020) confirming that overweight and obesity continues to rise in Ghana as the years go by.

Compared to a recent meta-analysis conducted in 2020 to evaluate childhood overweight and obesity in Ghana [37], our study points out that overweight and obesity is not just high among adults in Ghana but it is markedly higher than prevalence levels among Ghanaian children which was estimated as 10.7% and 8.6% respectively.

We also found that within the general Ghanaian population, females and urban dwellers tend to have higher BMI than males and people living in rural areas. Similarly, overweight and obesity are disproportionately distributed among adults in Ghana based on gender and whether they live in rural or urban area with females and urban dwellers worse affected. These findings are consistent with other studies conducted recently in Ghana, within the Africa region and globally. An earlier study in Ghana [18] found women to have a 1.3 times and 3.7 times higher prevalence of overweight and obesity respectively than men with urban dwellers registering a prevalence rates of 10.5% and 12.6% higher for overweight and obesity compared to rural dwellers. In the study conducted in Nigeria [38], pooled data from studies conducted between 2010 and 2022 was used in a meta-analysis and the prevalence of overweight was higher among women by 2% than men and obesity prevalence among women was 12.1% higher than in men. Ekpor et al. [39] studied overweight and obesity prevalence in the larger

Africa region and noted that females were more likely to be overweight and obese compared to their male counterparts. They also found that people living in urban areas had higher odds of being overweight and obese.

This disproportionately higher prevalence of overweight and obesity among women can be attributed to biological and sociological factors which tend to expose women more. Biologically, hormonal fluctuations within the lifespan of females may explain the increased risk of overweight and obesity [40]. Pregnancy is such an example of periods of hormonal fluctuations where women gain excessive weight and retention of same following pregnancy thus making it a significant factor in the development of overweight and obesity. Also, the differences in body composition and fat mass distribution between men and women make women more susceptible to overweight and obesity [41]. Sociocultural factors may also explain the apparent higher overweight and obesity prevalence rates among women in Africa and particularly Ghana. Men in Ghana tend to take up jobs that are more physically demanding whiles women mostly take up jobs that are predominantly sedentary in nature such as trade [42] thus contributing to disparities in physical activity levels [43]. Additionally, cultural glorification of women based on weight could also be linked to the high susceptibility of women to overweight and obesity. This was evidenced in a qualitative study conducted by Aryeetey et al, [44] in Accra, where they conclude that there is admiration for weight gain among women as it is perceived to be a sign of wealthy living and for married women it shows they are being taken good care of by their spouses. Urban areas are witnessing higher prevalence rates of overweight and obesity than rural areas as suggested by this study and the other referenced studies possibly due to several factors including the rapidly transforming transportation systems and food environment in urban areas in Ghana. Dake et al., [45] found that the local food environments in the urban poor areas in Ghana are suggestive of an obesogenic food environment dominated by fast food joints and convenience stores with significant associations with increased risk of obesity.

The current level of overweight and obesity established by this study and the disparities in these levels based on gender and urbanization suggests that Ghana continues to be in Stage 1 of the obesity transition as proposed by Jaacks et al. [46] which is characterized by a rise in obesity prevalence to above 5% but less than 20% among adults with comparably lower prevalence of childhood obesity and distinctly higher prevalence in females than males. The increasing prevalence in overweight and obesity especially in recent years as depicted by this study is corroborated by projections by the 2022 Global Nutrition Report [47] which indicates a continuous rising trajectory in obesity prevalence in Ghana with a narrowing of the gender disparities which will soon plunge Ghana into Stage 2 of the obesity transition.

SSA countries including Ghana still battle with a high burden of infectious diseases such as malaria and HIV/AIDs [48, 49]. At the same time prevalence of stunting and underweight in Ghana are 11% and 9% [17] respectively signifying that under nutrition still poses a significant threat to the health and wellbeing of vulnerable groups. These, together with the rising obesity situation and its associated risk of NCDs such as diabetes, hypertension and cancers create a complex milieu of a multiple burden of diseases in the Ghanaian population requiring a wide range of priority actions that will be all-encompassing and implemented at national scale.

Although it is appropriate to use national prevalence rates for obesity to recommend national policies, it is essential to consider sub national prevalence because of existential geographical differences in obesity prevalence which is mostly as a result of disparities in socioeconomic factors. This review noted that overweight and obesity prevalence varied between the southern, middle and northern sectors of Ghana. Higher prevalence rates of overweight and obesity were seen in the southern and middle belts whiles the northern belt of Ghana had lower prevalence. This difference seems to mimic the level of urbanization in these geographical locations of Ghana as published by Songsore [50] where Greater Accra which is the nucleus

of southern Ghana has 87.4% of its population in urban areas followed by the Ashanti region which is also the core of the middle belt with 53.2% urban inhabitants whiles urbanization in the rest of the regions were below the national average of 43.9% with the far northern regions; Upper West and Upper East having an urbanized population of 17.1% and 15.1% respectively. Evidence also points to an uneven distribution of wealth in Ghana [51, 52] where the northern regions are poorer than the rest of the country with the wealthier regions being the southern regions and middle sector regions. Although this review did not analyse prevalence rates by wealth status, it can be inferred that the sub national differences in obesity and overweight prevalence also reflects the influence of their wealth status.

## Study limitations

While this study presents the most current estimates of overweight and obesity prevalence in Ghana based on a high number of included studies and a large combined sample size, it is bedevilled with a number bottlenecks. There was a lack of national representativeness of studies that were used in the pooled estimates. Though the country was grouped geographically into three zones; southern, middle and northern zones, there were more studies in a few regions. For instance, the southern zone was dominated by studies from the Greater Accra region with a few from the Volta region. Similarly most studies in the middle zone were done in the Ashanti region whiles the northern zone had majority of the studies from the Northern region and a few from the Upper East region. Although this study estimates prevalence within the past 10 years we concede that many (about 70%) of the included studies were conducted between 2013 and 2016 which has the tendency to mask the current situation. An assessement of the quality of included studies also showed that many studies did not report on non-responders thus non-response bias could not be assessed to guarantee the external validity of the included studies.

There were also some limitations as well from the data analysis. In virtually all cases except for studies that were included in determining the pooled mean BMI, we rejected the null hypothesis that the true effect sizes were the same in all the studies signifying very high level of heterogeneity in the studies. In some of the sub groups such as age group, educational level and diabetes status, there were few included studies which could reduce the robustness of the random effects statistical model to estimate tau-square; which is the between study variance [53].

Irrespective of these limitations, the prevalence rates for overweight and obesity among adults in Ghana as determined by this study should be closely reflective of the current situation or more precisely the situation within the past 10 years.

## Conclusion

This systematic review and meta-analysis reveal that overweight and obesity prevalence is high among adults in Ghana with a sharp increase in recent years and existential disparities in the prevalence rates based on socio-demographic factors. This current level of overweight and obesity is a pointer to a potential worsening of the cardiovascular health outcomes of the population. More attention needs to be given to the promotion of the consumption of healthier diets which is supported with policy interventions such as regulation of the food industry as well as the promotion of increased physical activity levels especially through proper urban planning that tackles sedentary lifestyles borne out of urbanization and changing modes of transport. Whiles at this, more research will be needed to further understand the drivers of the rising overweight and obesity situation in Ghana.

## Supporting information

**S1 Checklist.**
(PDF)

**S1 Data.**
(XLSX)

## Author Contributions

**Conceptualization:** Mustapha Titi Yussif, Reginald Adjetey Annan.

**Formal analysis:** Mustapha Titi Yussif.

**Methodology:** Mustapha Titi Yussif, Araba Egyirba Morrison.

**Supervision:** Reginald Adjetey Annan.

**Writing – original draft:** Mustapha Titi Yussif.

**Writing – review & editing:** Araba Egyirba Morrison, Reginald Adjetey Annan.

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
