## [Decision Letter · Decision Letter 0]

6 Nov 2023

PGPH-D-23-01875

10-YEAR LEVEL, TRENDS AND SOCIO-DEMOGRAPHIC DISPARITIES OF OBESITY AMONG GHANAIAN ADULTS - A SYSTEMATIC REVIEW AND META-ANALYSIS OF OBSERVATIONAL STUDIES

Dear Dr. Yussif,

Thank you for submitting your manuscript to PLOS Global Public Health. After careful consideration, we feel that it has merit but does not fully meet PLOS Global Public Health’s publication criteria as it currently stands. Therefore, we invite you to submit a revised version of the manuscript that addresses the points raised during the review process.

To strengthen the robustness and clarity of your study, please implement the following changes:

It is imperative for the validity of your systematic review and meta-analysis that you clearly articulate compliance with the COSMOS-E and PRISMA guidelines. Please include a detailed statement in the methodology section confirming adherence to these guidelines and ensure that all the relevant criteria are met. Reference: Dekkers, O.M., Vandenbroucke, J.P., Cevallos, M., Renehan, A.G., Altman, D.G. and Egger, M., 2019. COSMOS-E: guidance on conducting systematic reviews and meta-analyses of observational studies of etiology. PLoS medicine, 16(2), p.e1002742.Your utilization of the AXIS tool for assessing quality and risk of bias is noted. However, the manuscript would benefit from a deeper explanation of how these assessments influenced your analysis. For instance, consider discussing the incorporation of quality scores within a quality effects model, as proposed by Doi and Thalib (2008). Doi SA, Thalib L. A quality-effects model for meta-analysis. Epidemiology. 2008 Jan;19(1):94-100. doi: 10.1097/EDE.0b013e31815c24e7. Erratum in: Epidemiology. 2010 Mar;21(2):278. PMID: 18090860.Screening and Duplicate Removal Documentation: The process of managing abstracts and eliminating duplicates is crucial in systematic reviews. Could you please elucidate the tools (such as Rayyan) or methods employed to this end? Moreover, details regarding the independent screening process and how discrepancies were resolved would greatly contribute to the transparency of your study's methodology.Assessment of Heterogeneity and Publication Bias: You've identified a high level of heterogeneity and publication bias within the included studies. To address this, we suggest exploring the use of the Doi plot and the Luis Furuya-Kanamori (LFK) index for a more robust visualization and quantification of asymmetry in study effects. Additionally, forest plots would be valuable for graphically presenting the meta-analysis results, including information on heterogeneity levels. The references provided by Furuya-Kanamori et al. (2018, 2021) offer a comprehensive guide on these methods.Furuya-Kanamori L, Doi SAR. LFK: Stata module to compute LFK index and Doi plot for detection of publication bias in meta-analysis. (2021) https://EconPapers.repec.org/RePEc:boc:bocode:s458762Furuya-Kanamori L, Barendregt JJ, Doi SAR. A new improved graphical and quantitative method for detecting bias in meta-analysis. JBI Evid Implement (2018) 16: https://journals.lww.com/ijebh/fulltext/2018/12000/a_new_improved_graphical_and_quantitative_method.3.aspx

A rebuttal letter that responds to each point raised by the editor and reviewer(s). You should upload this letter as a separate file labeled 'Response to Reviewers'.A marked-up copy of your manuscript that highlights changes made to the original version. You should upload this as a separate file labeled 'Revised Manuscript with Track Changes'.An unmarked version of your revised paper without tracked changes. You should upload this as a separate file labeled 'Manuscript'.Guidelines for resubmitting your figure files are available below the reviewer comments at the end of this letter.

We look forward to receiving your revised manuscript.

Kind regards,

Giridhara R Babu, MBBS, MPH, PhD

Academic Editor

Journal Requirements:

1. Please provide separate figure files in .tif or .eps format only and remove any figures embedded in your manuscript file. Please also ensure that all files are under our size limit of 10MB.

https://journals.plos.org/sustainabilitytransformation/s/figures 

https://journals.plos.org/sustainabilitytransformation/s/figures#loc-file-requirements

2. We have noticed that you have uploaded Supporting Information files, but you have not included a list of legends. Please add a full list of legends for your Supporting Information files after the references list.

3. In the online submission form, you indicated that "Data will be made available upon request to the corresponding author". All PLOS journals now require all data underlying the findings described in their manuscript to be freely available to other researchers, either 1. In a public repository, 2. Within the manuscript itself, or 3. Uploaded as supplementary information.

Additional Editor Comments (if provided):

Reviewers' comments:

Reviewer's Responses to Questions

**Comments to the Author**

1. Does this manuscript meet PLOS Global Public Health’s publication criteria? Is the manuscript technically sound, and do the data support the conclusions? The manuscript must describe methodologically and ethically rigorous research with conclusions that are appropriately drawn based on the data presented.

Reviewer #1: Yes

Reviewer #2: Yes

Reviewer #3: Yes

2. Has the statistical analysis been performed appropriately and rigorously?

Reviewer #1: Yes

Reviewer #2: Yes

Reviewer #3: Yes

3. Have the authors made all data underlying the findings in their manuscript fully available (please refer to the Data Availability Statement at the start of the manuscript PDF file)?

Reviewer #1: Yes

Reviewer #2: No

Reviewer #3: No

4. Is the manuscript presented in an intelligible fashion and written in standard English?

Reviewer #1: Yes

Reviewer #2: Yes

Reviewer #3: Yes

5. Review Comments to the Author

Reviewer #1: 1) Congratulations to the study for a great job done with this systematic review on the prevalence of obesity and overweight in Ghana.

2) Since the search duration was updated in January 2023, please correct the search period to Oct 2022 to Jan 2023.

3) Line 129: why were other studies included irrespective of the metabolic status of the participants? What value addition or relevant information were gleaned from these other studies? If line 135 is making reference to the studies referred to on line 129, then please rephrase to make the statement clearer that relevant information in these studies would be used to compute prevalence.

4) Lines 211 and 212: can pass as a limitation. There should be a period at the end of line 212.

5) Line 256: Table 2 didn’t show any level of significance for the variable “marital status”

6) Figure 11: What could have accounted for the change in trends for the two variables in 2017/18 especially being overweight?

Reviewer #2: Comments

This article is a good report as a comprehensive systematic review and meta-analysis. Still, there needs to be a revision in some parts.

Method:

1. Please indicate whether the authors followed the PRISMA guideline.

2. Please attach all search terms used in each literature portal.

Analysis:

1. The Statistical values of publication bias analysis need to be in the Funnel plot figures or result tables.

2. Please add a Table for the result of Proportion subgroup analysis (Variation in Obesity Proportion according to subgroups)

3. Please add a Figure for "overweight and obesity diabetes patients in Ghana."

4. Figure 12 is empty. Please be sure to see Figure 12 again.

Reviewer #3: The is an excellent and informative manuscript which can help policy and decision makers in Ghana to craft appropriate strategies and interventions to address the problems; which are likely contributing to raising non-communicable diseases (NCDs) in Ghana just as in other low-income and middle-income countries (LMICs). There are very few minor areas which I suggest to the Authors to address them as follows:

ABSTRACT

Line 35: needs corrections on the 25.6kb-2 by adding “m” so that it will be 25.6kgm-2

INTRODUCTION

Line 51: since in Line 53 the authors have used SSA to mean Sub-Saharan Africa; they need to insert the abbreviation SSA in brackets in line 51 just after Africa to be as follows: Sub-Saharan Africa (SSA) has seen more…...

Line 67: I suggest to the authors to replace “low- and middle- income countries (LMICs) with its abbreviation LMICs since it was already abbreviated in Line 50.

Line 93: (Meanwhile, the prevalence of overweight and obesity continuous to rise in Ghana over the), I hope the authors meant “continues” instead of “continuous”.

METHODS

Search Strategy

Line 110: I suggest to the authors to edit “PUBMED” to be “PubMed”.

RESULTS

Overweight and obesity among diabetes patients in Ghana

Line 354: I suggest to the authors to replace “cardiovascular diseases” with its abbreviation “CVDs”.

DISCUSSION

Lines 377–378: I suggest to the authors to replace “sub-Saharan Africa” with its abbreviation SSA used in Lines 51 & 53.

Line 384: (years (2019/2020) confirming that overweight and obesity continuous to rise in Ghana as the), I hope the authors meant “continues” instead of “continuous”.

Line 429: (levels based on gender and urbanization suggests that Ghana continuous to be in Stage 1 of the), I hope the authors meant “continues” instead of “continuous”.

Line 437: I suggest to the authors to replace “Sub Saharan Africa” with its abbreviation SSA.

Line 441: I suggest to the authors to replace “non-communicable diseases” with its abbreviation NCDs which was already abbreviated in Line 54.

Line 458: (poorer than the rest of rest of the country with the wealthier regions being the southern regions), The authors need to DELETE the repeating words “rest of”.

FIGURE 12: socio-demographic disparities in obesity prevalence in Ghana

The figure seems to be incomplete in terms of displaying the information on disparities. Therefore, I suggest to the authors to recheck it; &

I also suggest to the authors to ensure that it (Fig.12) is linked in the text to enable a reader to follow through easily.

6. PLOS authors have the option to publish the peer review history of their article (what does this mean?). If published, this will include your full peer review and any attached files.

**Do you want your identity to be public for this peer review?** For information about this choice, including consent withdrawal, please see our Privacy Policy.

Reviewer #1: No

Reviewer #2: No

Reviewer #3: **Yes: **Eliudi Saria Eliakimu

---

## [Decision Letter · Decision Letter 1]

4 Jan 2024

10-YEAR LEVEL, TRENDS AND SOCIO-DEMOGRAPHIC DISPARITIES OF OBESITY AMONG GHANAIAN ADULTS - A SYSTEMATIC REVIEW AND META-ANALYSIS OF OBSERVATIONAL STUDIES

PGPH-D-23-01875R1

Dear Mr Yussif,

We are pleased to inform you that your manuscript '10-YEAR LEVEL, TRENDS AND SOCIO-DEMOGRAPHIC DISPARITIES OF OBESITY AMONG GHANAIAN ADULTS - A SYSTEMATIC REVIEW AND META-ANALYSIS OF OBSERVATIONAL STUDIES' has been provisionally accepted for publication in PLOS Global Public Health.

Best regards,

Giridhara R Babu, MBBS, MPH, PhD

Academic Editor

The reviewers have recommended to accept this article. 

Reviewer Comments (if any, and for reference):

Reviewer's Responses to Questions

**Comments to the Author**

1. If the authors have adequately addressed your comments raised in a previous round of review and you feel that this manuscript is now acceptable for publication, you may indicate that here to bypass the “Comments to the Author” section, enter your conflict of interest statement in the “Confidential to Editor” section, and submit your "Accept" recommendation.

Reviewer #1: All comments have been addressed

Reviewer #2: All comments have been addressed

Reviewer #3: All comments have been addressed

2. Does this manuscript meet PLOS Global Public Health’s publication criteria? Is the manuscript technically sound, and do the data support the conclusions? The manuscript must describe methodologically and ethically rigorous research with conclusions that are appropriately drawn based on the data presented.

Reviewer #1: Yes

Reviewer #2: Yes

Reviewer #3: Yes

3. Has the statistical analysis been performed appropriately and rigorously?

Reviewer #1: Yes

Reviewer #2: Yes

Reviewer #3: Yes

4. Have the authors made all data underlying the findings in their manuscript fully available (please refer to the Data Availability Statement at the start of the manuscript PDF file)?

Reviewer #1: Yes

Reviewer #2: Yes

Reviewer #3: Yes

5. Is the manuscript presented in an intelligible fashion and written in standard English?

Reviewer #1: Yes

Reviewer #2: Yes

Reviewer #3: Yes

6. Review Comments to the Author

Reviewer #1: Congratulations to the research team.

Reviewer #2: (No Response)

Reviewer #3: (No Response)

7. PLOS authors have the option to publish the peer review history of their article (what does this mean?). If published, this will include your full peer review and any attached files.

**Do you want your identity to be public for this peer review?** For information about this choice, including consent withdrawal, please see our Privacy Policy.

Reviewer #1: No

Reviewer #2: No

Reviewer #3: **Yes: **Eliudi Saria Eliakimu
